# SCALEFORMER: ITERATIVE MULTI-SCALE REFINING TRANSFORMERS FOR TIME SERIES FORECASTING

**Mohammad Amin Shabani,**
Simon Fraser University, Canada
& Borealis AI, Canada
mshabani@sfu.ca

**Amir Abdi, Lili Meng, Tristan Sylvain**
Borealis AI, Canada
{firstname.lastname}@borealisai.com

## ABSTRACT

The performance of time series forecasting has recently been greatly improved by the introduction of transformers. In this paper, we propose a general multi-scale framework that can be applied to the state-of-the-art transformer-based time series forecasting models (FEDformer, Autoformer, etc.). By iteratively refining a forecasted time series at multiple scales with shared weights, introducing architecture adaptations, and a specially-designed normalization scheme, we are able to achieve significant performance improvements, from $5.5\%$ to $38.5\%$ across datasets and transformer architectures, with minimal additional computational overhead. Via detailed ablation studies, we demonstrate the effectiveness of each of our contributions across the architecture and methodology. Furthermore, our experiments on various public datasets demonstrate that the proposed improvements outperform their corresponding baseline counterparts. Our code is publicly available in https://github.com/BorealisAI/scaleformer.

## 1 INTRODUCTION

Integrating information at different time scales is essential to accurately model and forecast time series (Mozer, 1991; Ferreira et al., 2006). From weather patterns that fluctuate both locally and globally, as well as throughout the day and across seasons and years, to radio carrier waves which contain relevant signals at different frequencies, time series forecasting models need to encourage *scale awareness* in learnt representations. While transformer-based architectures have become the mainstream and state-of-the-art for time series forecasting in recent years, advances have focused mainly on mitigating the standard quadratic complexity in time and space, e.g., attention (Li et al., 2019; Zhou et al., 2021) or structural changes (Xu et al., 2021; Zhou et al., 2022b), rather than explicit scale-awareness. The essential cross-scale feature relationships are often learnt implicitly, and are not encouraged by architectural priors of any kind beyond the stacked attention blocks that characterize the transformer models. Autoformer (Xu et al., 2021) and Fedformer (Zhou et al., 2022b) introduced some emphasis on scale-awareness by enforcing different computational paths for the trend and seasonal components of the input time series; however, this structural prior only focused on two scales: low- and high-frequency components. **Given their importance to forecasting, can we make transformers more scale-aware?**

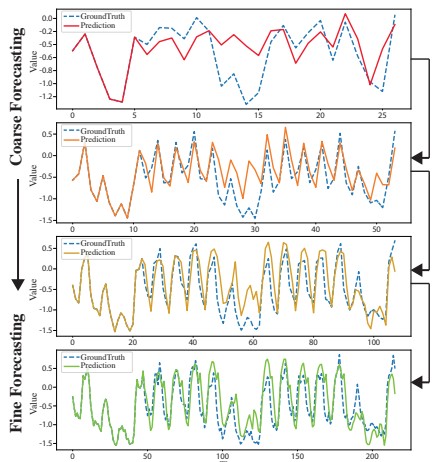

Figure 1: Intermediate forecasts by our model at different time scales. Iterative refinement of a time series forecast is a strong structural prior that benefits time series forecasting.

We enable this scale-awareness with *Scaleformer*. In our proposed approach, showcased in Figure 1, time series forecasts are iteratively refined at successive time-steps, allowing the model to better capture the inter-dependencies and specificities of each scale. However, scale itself is not sufficient. Iterative refinement at different scales can cause significant distribution shifts between intermediate

forecasts which can lead to runaway error propagation. To mitigate this issue, we introduce cross-scale normalization at each step.

Our approach re-orders model capacity to shift the focus on scale awareness, but does not fundamentally alter the attention-driven paradigm of transformers. As a result, it can be readily adapted to work jointly with multiple recent time series transformer architectures, acting broadly orthogonally to their own contributions. Leveraging this, we chose to operate with various transformer-based backbones (e.g. Fedformer, Autoformer, Informer, Reformer, Performer) to further probe the effect of our multi-scale method on a variety of experimental setups.

Our contributions are as follows: (1) we introduce a novel iterative scale-refinement paradigm that can be readily adapted to a variety of transformer-based time series forecasting architectures. (2) To minimize distribution shifts between scales and windows, we introduce cross-scale normalization on outputs of the Transformer. (3) Using Informer and AutoFormer, two state-of-the-art architectures, as backbones, we demonstrate empirically the effectiveness of our approach on a variety of datasets. Depending on the choice of transformer architecture, our mutli-scale framework results in mean squared error reductions ranging from $5.5\%$ to $38.5\%$. (4) Via a detailed ablation study of our findings, we demonstrate the validity of our architectural and methodological choices.

## 2 RELATED WORKS

**Time-series forecasting**: Time-series forecasting plays an important role in many domains, including: weather forecasting (Murphy, 1993), inventory planning (Syntetos et al., 2009), astronomy (Scargle, 1981), economic and financial forecasting (Krollner et al., 2010). One of the specificities of time series data is the need to capture *seasonal* trends (Brockwell & Davis, 2009). There exits a vast variety of time-series forecasting models (Box & Jenkins, 1968; Hyndman et al., 2008; Salinas et al., 2020; Rangapuram et al., 2018; Bai et al., 2018; Wu et al., 2020). Early approaches such as ARIMA (Box & Jenkins, 1968) and exponential smoothing models (Hyndman et al., 2008) were followed by the introduction of neural network based approaches involving either Recurrent Neural Netowkrs (RNNs) and their variants (Salinas et al., 2020; Rangapuram et al., 2018; Salinas et al., 2020) or Temporal Convolutional Networks (TCNs) (Bai et al., 2018).

More recently, time-series Transformers (Wu et al., 2020; Zerveas et al., 2021; Tang & Matteson, 2021) were introduced for the forecasting task by leveraging self-attention mechanisms to learn complex patterns and dynamics from time series data. Informer (Zhou et al., 2021) reduced quadratic complexity in time and memory to $O(L \log L)$ by enforcing sparsity in the attention mechanism with the ProbSparse attention. Yformer (Madhusudhanan et al., 2021) proposed a Y-shaped encoder-decoder architecture to take advantage of the multi-resolution embeddings. Autoformer (Xu et al., 2021) used a cross-correlation-based attention mechanism to operate at the level of subsequences. FEDformer (Zhou et al., 2022b) employs frequency transform to decompose the sequence into multiple frequency domain modes to extract the feature, further improving the performance of Autoformer.

**Multi-scale neural architectures**: Multi-scale and hierarchical processing is useful in many domains, such as computer vision (Fan et al., 2021; Zhang et al., 2021; Liu et al., 2018), natural language processing (Nawrot et al., 2021; Subramanian et al., 2020; Zhao et al., 2021) and time series forecasting (Chen et al., 2022; Ding et al., 2020). Multiscale Vision Transformers (Fan et al., 2021) is proposed for video and image recognition, by connecting the seminal idea of multiscale feature hierarchies with transformer models, however, it focuses on the spatial domain, specially designed for computer vision tasks. Cui et al. (2016) proposed to use different transformations of a time series such as downsampling and smoothing in parallel to the original signal to better capture temporal patterns and reduce the effect of random noise. Many different architectures have been proposed recently (Chung et al., 2016; Che et al., 2018; Shen et al., 2020; Chen et al., 2021) to improve RNNs in tasks such as language processing, computer vision, time-series analysis, and speech recognition. However, these methods are mainly focused on proposing a new RNN-based module which is not applicable to transformers directly. The same direction has been also investigated in Transformers, TCN, and MLP models. Recent work Du et al. (2022) proposed multi-scale segment-wise correlations as a multi-scale version of the self-attention mechanism. Our work is orthogonal to the above methods

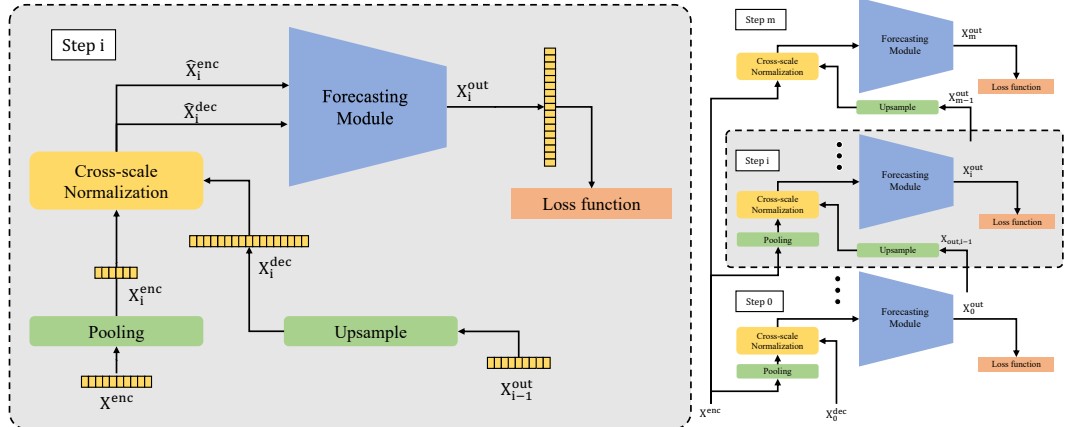

Figure 2: Overview of the proposed *Scaleformer* framework. (Left) Representation of a single scale block. In each step, we pass the normalized upsampled output from previous step along with the normalized downsampled encoder as the input. (Right) Representation of the full architecture. We process the input in a multi-scale manner iteratively from the smallest scale to the original scale.

as a model-agnostic framework to utilize multi-scale time-series in transformers while keeping the number of parameters and time complexity roughly the same.

# 3 METHOD

In this section, we first introduce the problem setting in Sec. 3.1, then describe the proposed framework in Sec. 3.2, and the normalization scheme in Sec. 3.3. We provide details on the input's representation in Sec. 3.4, and the loss function in Sec. 3.5.

## 3.1 PROBLEM SETTING

We denote $\mathbf{X}^{(L)}$ and $\mathbf{X}^{(H)}$ the look-back and horizon windows for the forecast, respectively, of corresponding lengths $\ell_L, \ell_H$. Given a starting time $t_0$ we can express these time-series of dimension $d_x$, as follows: $\mathbf{X}^{(L)} = \{\mathbf{x}_t | \mathbf{x}_t \in \mathbb{R}^{d_x}, t \in [t_0, t_0 + \ell_L]\}$ and $\mathbf{X}^{(H)} = \{\mathbf{x}_t | \mathbf{x}_t \in \mathbb{R}^{d_x}, t \in [t_0 + \ell_L + 1, t_0 + \ell_L + \ell_H]\}$. The goal of the forecasting task is to predict the horizon window $\mathbf{X}^{(H)}$ given the look-back window $\mathbf{X}^{(L)}$.

## 3.2 MULTI-SCALE FRAMEWORK

Our proposed framework applies successive transformer modules to iteratively refine a time-series forecast, at different temporal scales. The proposed framework is shown in Figure 2.

Given an input time-series $\mathbf{X}^{(L)}$, we iteratively apply the same neural module mutliple times at different temporal scales. Concretely, we consider a set of scales $S = \{s^m, ..., s^2, s^1, 1\}$ (i.e. for the default scale of $s = 2$, $S$ is a set of consecutive powers of 2), where $m = \lfloor \log_s \ell_L \rfloor - 1$ and $s$ is a downscaling factor. The input to the encoder at the $i$-th step ($0 \leq i \leq m$) is the original look-back window $\mathbf{X}^{(L)}$, downsampled by a scale factor of $s_i \equiv s^{m-i}$ via an average pooling operation. The input to the decoder, on the other hand, is $\mathbf{X}_{i-1}^{\text{out}}$ upsampled by a factor of $s$ via a linear interpolation.

Finally, $\mathbf{X}_0^{\text{dec}}$ is initialized to an array of 0s. The model performs the following operations:

$$\mathbf{x}_{t,i} = \frac{1}{s_i} \sum_{\tau+1}^{\tau+s_i} \mathbf{x}_\tau, \quad \tau = t \times s_i \tag{1}$$

$$\mathbf{X}_i^{(L)} = \left\{ \mathbf{x}_{t,i} \ \middle| \ \frac{t_0}{s_i} \leq t \leq \frac{t_0 + \ell_L}{s_i} \right\} \tag{2}$$

$$\mathbf{X}_i^{(H)} = \left\{ \mathbf{x}_{t,i} \ \middle| \ \frac{t_0 + \ell_L + 1}{s_i} \leq t \leq \frac{t_0 + \ell_L + \ell_H}{s_i} \right\}, \tag{3}$$

where $\mathbf{X}_i^{(L)}$ and $\mathbf{X}_i^{(H)}$ are the look-back and horizon windows at the $i$th step at time t with the scale factor of $s^{m-i}$ and with the lengths of $\ell_{L,i}$ and $\ell_{H,i}$, respectively. Assuming $\mathbf{x}'_{t,i-1}$ is the output of the forecasting module at step $i-1$ and time $t$, we can define $\mathbf{X}_i^{\text{enc}}$ and $\mathbf{X}_i^{\text{dec}}$ as the inputs to the normalization:

$$\mathbf{X}_i^{\text{enc}} = \mathbf{X}_i^{(L)} \tag{4}$$

$$\mathbf{x}''_{t,i} = \mathbf{x}'_{\lfloor t/s \rfloor, i-1} + \left( \mathbf{x}'_{\lceil t/s \rceil, i-1} - \mathbf{x}'_{\lfloor t/s \rfloor, i-1} \right) \times \frac{t - \lfloor t/s \rfloor}{s} \tag{5}$$

$$\mathbf{X}_i^{\text{dec}} = \left\{ \mathbf{x}''_{t,i} \ \middle| \ \frac{t_0 + \ell_L + 1}{s_i} \leq t \leq \frac{t_0 + \ell_L + \ell_H}{s_i} \right\}. \tag{6}$$

Finally, we calculate the error between $\mathbf{X}_i^{(H)}$ and $\mathbf{X}_i^{\text{out}}$ as the loss function to train the model. Please refer to Algorithm 1 for details on the sequence of operations performed during the forward pass.

---

**Algorithm 1** Scaleformer: Iterative Multi-scale Refining Transformer

---

**Require:** input lookback window $\mathbf{X}^{(L)} \in \mathbb{R}^{\ell_L \times d_x}$, scale factor $s$,
    a set of scales $S = \{s^m, ..., s^2, s^1, 1\}$, Horizon length $\ell_H$, and Transformer module $F$.
  **for** $i \leftarrow 0$ to $m$ **do**
    $\mathbf{X}_i^{\text{enc}} \leftarrow \text{AvgPool}\left( \mathbf{X}^{(L)}, \text{window\_size=}s^{m-i} \right)$           ▷ Equation (1) and (2) of the paper
    **if** $i = 0$ **then**
      $\mathbf{X}_i^{\text{dec}} \leftarrow \left[ \overrightarrow{0}^{\ell_H} \right]$
    **else**
      $\mathbf{X}_i^{\text{dec}} \leftarrow \text{Upsample}\left( \mathbf{X}_{i-1}^{\text{out}}, \text{scale=}s \right)$         ▷ Equation (5) and (6) of the paper
    **end if**
    $\bar{\mu}_{\mathbf{X}_i} \leftarrow \frac{1}{\ell_{L,i} + \ell_{H,i}} \left( \sum_{\mathbf{x}^{\text{enc}} \in \mathbf{X}_i^{\text{enc}}} \mathbf{x}^{\text{enc}} + \sum_{\mathbf{x}^{\text{dec}} \in \mathbf{X}_i^{\text{dec}}} \mathbf{x}^{\text{dec}} \right)$
    $\hat{\mathbf{X}}_i^{\text{dec}} \leftarrow \mathbf{X}_i^{\text{dec}} - \bar{\mu}_{\mathbf{X}_i}$
    $\hat{\mathbf{X}}_i^{\text{enc}} \leftarrow \mathbf{X}_i^{\text{enc}} - \bar{\mu}_{\mathbf{X}_i}$
    $\mathbf{X}_i^{\text{out}} \leftarrow F\left( \mathbf{X}_i^{\text{enc}}, \mathbf{X}_i^{\text{dec}} \right) + \bar{\mu}_{\mathbf{X}_i}$
  **end for**
**Ensure:** $\mathbf{X}_{:}^{\text{out}}$                     ▷ return the prediction at all scales

---

## 3.3 Cross-scale Normalization

Given a set of input series $(\mathbf{X}_i^{\text{enc}}, \mathbf{X}_i^{\text{dec}})$, with dimensions $\ell_{L_i} \times d_x$ and $\ell_{H_i} \times d_x$, respectively for the encoder and the decoder of the transformer in $i$th step, we normalize each series based on the temporal average of $\mathbf{X}_i^{\text{enc}}$ and $\mathbf{X}_i^{\text{dec}}$. More formally:

$$\bar{\mu}_{\mathbf{X}_i} = \frac{1}{\ell_{L,i} + \ell_{H,i}} \left( \sum_{\mathbf{x}^{\text{enc}} \in \mathbf{X}_i^{\text{enc}}} \mathbf{x}^{\text{enc}} + \sum_{\mathbf{x}^{\text{dec}} \in \mathbf{X}_i^{\text{dec}}} \mathbf{x}^{\text{dec}} \right) \tag{7}$$

$$\hat{\mathbf{X}}_i^{\text{dec}} = \mathbf{X}_i^{\text{dec}} - \bar{\mu}_{\mathbf{X}_i}, \qquad \hat{\mathbf{X}}_i^{\text{enc}} = \mathbf{X}_i^{\text{enc}} - \bar{\mu}_{\mathbf{X}_i} \tag{8}$$

where $\bar{\mu}_{\mathbf{X}_i} \in \mathbb{R}^{d_x}$ is the average over the temporal dimension of the concatenation of both look-back window and the horizon. Here, $\hat{\mathbf{X}}_i^{\text{enc}}$ and $\hat{\mathbf{X}}_i^{\text{dec}}$ are the inputs of the $i$th step to the forecasting module.

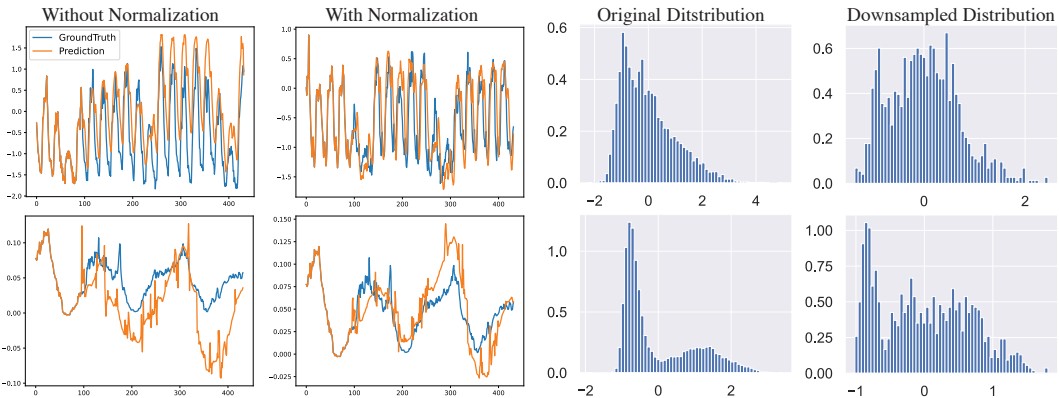

Figure 3: The figure shows the output results of two series using the same trained multi-scale model with and without shifting the data (left) which demonstrates the importance of normalization. On the right, we can see the distribution changes due to the downsampling of two series compared to the original scales from the Electricity dataset.

Distribution shift is when the distribution of input to a model or its sub-components changes across training to deployment (Shimodaira, 2000; Ioffe & Szegedy, 2015). In our context, two distinct distribution shifts are common. First, there is a natural distribution shift between the look-back window and the forecast window (the covariate shift). Additionally, there is a distribution shift between the predicted forecast windows at two consecutive scales which is a result of the upsampling operation alongside the error accumulation during the intermediate computations. As a result, normalizing the output at a given step by either the look-back window statistics or the previously predicted forecast window statistics result in an accumulation of errors across steps. We mitigate this by considering a moving average of forecast and look-back statistics as the basis for the output normalization. While this change might appear relatively minor, it has a significant impact on the resulting distribution of outputs. The improvement is more evident when compared to the alternative approaches, namely, normalizing by either look-back or previous forecast window statistics.

## 3.4 INPUT EMBEDDING

Following the previous works, we embed our input to have the same number of features as the hidden dimension of the model. The embedding consists of three parts: (1) *Value embedding* which uses a linear layer to map the input observations of each step $\mathbf{x_t}$ to the same dimension as the model. We further concatenate an additional value $0, 0.5$, or $1$ respectively showing if each observation is coming from the look-back window, zero initialization, or the prediction of the previous steps. (2) *Temporal Embedding* which again uses a linear layer to embed the time stamp related to each observation to the hidden dimension of the model. Here we concatenate an additional value $1/s_i - 0.5$ as the current scale for the network before passing to the linear layer. (3) We also use a fixed *positional embedding* which is adapted to the different scales $s_i$ as follows:

$$PE(\text{pos}, 2k, s_i) = \sin\left(\frac{\text{pos} \times s_i}{10000^{2k/d_{\text{model}}}}\right), \quad PE(\text{pos}, 2k+1, s_i) = \cos\left(\frac{\text{pos} \times s_i}{10000^{2k/d_{\text{model}}}}\right) \quad (9)$$

## 3.5 LOSS FUNCTION

Using the standard MSE objective to train time-series forecasting models leaves them sensitive to outliers. One possible solution is to use objectives more robust to outliers, such as the Huber loss (Huber, 1964). However, when there are no major outliers, such objectives tend to underperform. Given the heterogeneous nature of the data, we instead utilize the adaptive loss (Barron, 2019):

$$f(\xi, \alpha, c) = \frac{|\alpha - 2|}{\alpha}\left(\left(\frac{(\xi/c)^2}{|\alpha - 2|} + 1\right)^{\alpha/2} - 1\right) \quad (10)$$

Table 1: Comparison of the MSE and MAE results for our proposed multi-scale framework version of different methods (-**MSA**) with respective baselines. Results are given in the *multi-variate* setting, for different lenghts of the horizon window. The best results are shown in **Bold**. Our method outperforms vanilla version of the baselines over almost all datasets and settings. The average improvement (error reduction) is shown in Green numbers at the bottom with respect the base models.

| Method/ Dataset | | FEDformer MSE | MAE | FED-**MSA** MSE | MAE | Autoformer MSE | MAE | Auto-**MSA** MSE | MAE | Informer MSE | MAE | Info-**MSA** MSE | MAE | Reformer MSE | MAE | Ref-**MSA** MSE | MAE | Performer MSE | MAE | Per-**MSA** MSE | MAE |
|---|---|---|---|---|---|---|---|---|---|---|---|---|---|---|---|---|---|---|---|---|---|
| Exchange | 96 | 0.155 | 0.285 | **0.109** | **0.240** | 0.154 | 0.285 | **0.126** | **0.259** | 0.966 | 0.792 | **0.168** | **0.298** | 1.063 | 0.826 | **0.182** | **0.311** | 0.667 | 0.669 | **0.179** | **0.305** |
| | 192 | 0.274 | 0.384 | **0.241** | **0.353** | 0.356 | 0.428 | **0.253** | **0.373** | 1.088 | 0.842 | **0.427** | **0.484** | 1.597 | 1.029 | **0.375** | **0.446** | 1.339 | 0.904 | **0.439** | **0.486** |
| | 336 | **0.452** | **0.498** | 0.471 | 0.508 | 0.441 | 0.495 | 0.519 | 0.538 | 1.598 | 1.016 | **0.500** | **0.535** | 1.712 | 1.070 | **0.605** | **0.591** | 1.081 | 0.844 | **0.563** | **0.577** |
| | 720 | **1.172** | **0.839** | 1.259 | 0.865 | 1.118 | 0.819 | **0.928** | **0.751** | 2.679 | 1.340 | **1.017** | **0.790** | 1.918 | 1.160 | **1.089** | **0.857** | 0.867 | 0.766 | 1.219 | 0.882 |
| Weather | 96 | 0.288 | 0.365 | **0.220** | **0.289** | 0.267 | 0.334 | **0.163** | **0.226** | 0.388 | 0.435 | **0.210** | **0.279** | 0.347 | 0.388 | **0.199** | **0.263** | 0.441 | 0.479 | **0.228** | **0.291** |
| | 192 | 0.368 | 0.425 | **0.341** | **0.385** | 0.323 | 0.376 | **0.221** | **0.290** | 0.433 | 0.453 | **0.289** | **0.333** | 0.463 | 0.469 | **0.294** | **0.355** | 0.475 | 0.501 | **0.302** | **0.357** |
| | 336 | **0.447** | 0.469 | 0.463 | **0.455** | 0.364 | 0.397 | **0.282** | **0.340** | 0.610 | 0.551 | **0.418** | **0.427** | 0.734 | 0.622 | **0.463** | **0.464** | 0.478 | 0.482 | **0.441** | **0.456** |
| | 720 | **0.640** | 0.574 | 0.682 | **0.565** | 0.425 | 0.434 | **0.369** | **0.396** | 0.978 | 0.723 | **0.595** | **0.532** | 0.815 | 0.674 | **0.493** | **0.471** | 0.563 | 0.552 | 0.817 | 0.655 |
| Electricity | 96 | 0.201 | 0.317 | **0.182** | **0.297** | 0.197 | 0.312 | **0.188** | **0.303** | 0.344 | 0.421 | **0.203** | **0.315** | 0.294 | 0.382 | **0.183** | **0.291** | 0.294 | 0.387 | **0.190** | **0.300** |
| | 192 | 0.200 | 0.314 | **0.188** | **0.300** | 0.219 | 0.329 | **0.197** | **0.310** | 0.344 | 0.426 | **0.219** | **0.331** | 0.331 | 0.409 | **0.194** | **0.304** | 0.305 | 0.400 | **0.200** | **0.310** |
| | 336 | 0.214 | 0.330 | **0.210** | **0.324** | 0.263 | 0.359 | **0.224** | **0.333** | 0.358 | 0.440 | **0.253** | **0.360** | 0.361 | 0.428 | **0.209** | **0.321** | 0.331 | 0.416 | **0.209** | **0.322** |
| | 720 | 0.239 | 0.350 | **0.232** | **0.339** | 0.290 | 0.380 | **0.249** | **0.358** | 0.386 | 0.452 | **0.293** | **0.390** | 0.316 | 0.393 | **0.234** | **0.340** | 0.304 | 0.336 | **0.228** | **0.335** |
| Traffic | 96 | 0.601 | 0.376 | **0.564** | **0.351** | 0.628 | 0.393 | **0.567** | **0.350** | 0.748 | 0.426 | **0.597** | **0.369** | 0.698 | 0.386 | **0.615** | **0.377** | 0.730 | 0.405 | **0.612** | **0.371** |
| | 192 | 0.603 | 0.379 | **0.570** | **0.349** | 0.634 | 0.401 | **0.589** | **0.360** | 0.772 | 0.436 | **0.655** | **0.399** | 0.694 | 0.378 | **0.613** | **0.367** | 0.698 | 0.387 | **0.608** | **0.368** |
| | 336 | 0.602 | 0.375 | **0.576** | **0.349** | 0.619 | 0.385 | **0.609** | **0.383** | 0.868 | 0.493 | **0.761** | **0.455** | 0.695 | 0.377 | **0.617** | **0.360** | 0.678 | 0.370 | **0.604** | **0.356** |
| | 720 | 0.615 | 0.378 | **0.602** | **0.360** | 0.656 | 0.403 | **0.642** | **0.397** | 1.074 | 0.606 | **0.924** | **0.521** | 0.692 | 0.376 | **0.638** | **0.360** | 0.672 | 0.364 | **0.634** | **0.360** |
| ILI | 24 | 3.025 | 1.189 | **2.745** | **1.075** | 3.862 | 1.370 | **3.370** | **1.213** | 5.402 | 1.581 | **3.742** | **1.252** | 3.961 | 1.289 | **3.534** | **1.212** | 4.806 | 1.471 | **3.437** | **1.148** |
| | 32 | 3.034 | 1.201 | **2.748** | **1.072** | 3.871 | 1.379 | **3.088** | **1.164** | 5.296 | 1.587 | **3.807** | **1.272** | 4.022 | 1.311 | **3.652** | **1.235** | 4.669 | 1.455 | **4.055** | **1.248** |
| | 48 | **2.444** | **1.041** | 2.793 | 1.059 | **2.891** | **1.138** | 3.207 | 1.153 | 5.226 | 1.569 | **3.940** | **1.272** | 4.269 | 1.340 | **3.506** | **1.168** | 4.488 | 1.371 | **4.055** | **1.248** |
| | 64 | 2.686 | 1.112 | **2.678** | **1.071** | 3.164 | 1.223 | **2.954** | **1.112** | 5.304 | 1.578 | **3.670** | **1.234** | 4.370 | 1.385 | **3.487** | **1.177** | 4.607 | 1.404 | **3.828** | **1.224** |
| **Vs Ours** | | | | 5.6% | 5.9% | | | 13.5% | 9.1% | | | 38.5% | 26.7% | | | 38.3% | 25.2% | | | 23.3% | 16.9% |

with $\xi = (\mathbf{X}_i^{\text{out}} - \mathbf{X}_i^{(H)})$ in step $i$. The parameters $\alpha$ and $c$, which modulate the loss sensitivity to outliers, are learnt in an end-to-end fashion during training. To the best of our knowledge, this is the first time this objective has been adapted to the context of time-series forecasting.

# 4 EXPERIMENTS

In this section, we first showcase the main results of our proposed approach on a variety of forecasting dataset in Sec. 4.1. Then, we provide an ablation study of the different components of our model in Sec. 4.2, and also present qualitative results in Sec. 4.3. Moreover, we discuss a series of additional extensions of our method in Sec. 4.4, shedding light on promising future directions.

## 4.1 MAIN RESULTS

**Baselines**: To measure the effectiveness of the proposed framework, we mainly use state-of-the-art transformer-based models FedFormer (Zhou et al., 2022b) Reformer (Kitaev et al., 2020), Performer (Choromanski et al., 2020), Informer (Zhou et al., 2021) and Autoformer (Xu et al., 2021) which are proven to have beaten other transformer-based (e.g. LogTrans (Li et al., 2019), Reformer (Kitaev et al., 2020)), RNN-based (e.g. LSTMNet (Lai et al., 2018), LSTM) and TCN (Bai et al., 2018) models. For brevity, we only keep comparisons with the transformer models in the tables.

**Datasets**: We consider four public datasets with different characteristics to evaluate our proposed framework. **Electricity Consuming Load (ECL)**[1] corresponds to the electricity consumption (Kwh) of 321 clients. **Traffic**[2] aggregates the hourly occupancy rate of 963 car lanes of San Francisco bay area freeways. **Weather**[3] contains 21 meteorological indicators, such as air temperature, humidity, etc, recorded every 10 minutes for the entirety of 2020. **Exchange-Rate** (Lai et al., 2018) collects the daily exchange rates of 8 countries (Australia, British, Canada, Switzerland, China, Japan, New Zealand and Singapore) from 1990 to 2016. **National Illness (ILI)** [4] corresponds to the weekly

---

[1]https://archive.ics.uci.edu/ml/datasets/ElectricityLoadDiagrams20112014

[2]https://pems.dot.ca.gov

[3]https://www.bgc-jena.mpg.de/wetter/

[4]https://gis.cdc.gov/grasp/fluview/fluportaldashboard.html

Table 2: Multi-scale framework without cross-scale normalization. Correctly normalizing across different scales (as per our cross-mean normalization) is essential to obtain good performance when using the multi-scale framework.

| Dataset | | FEDformer | | FED-MS (w/o N) | | Autoformer | | Auto-MS (w/o N) | | Informer | | Info-MS (w/o N) | |
|---|---|---|---|---|---|---|---|---|---|---|---|---|---|
| Metric | | MSE | MAE | MSE | MAE | MSE | MAE | MSE | MAE | MSE | MAE | MSE | MAE |
| Weather | 96 | **0.288** | 0.365 | 0.300 | **0.342** | 0.267 | 0.334 | **0.191** | **0.277** | 0.388 | 0.435 | 0.402 | 0.438 |
| | 192 | **0.368** | 0.425 | 0.424 | **0.422** | 0.323 | 0.376 | **0.281** | **0.360** | 0.433 | 0.453 | **0.393** | **0.434** |
| | 336 | **0.447** | **0.469** | 0.531 | 0.493 | **0.364** | **0.397** | 0.376 | 0.420 | 0.610 | 0.551 | **0.566** | **0.528** |
| | 720 | **0.640** | **0.574** | 0.714 | 0.576 | **0.425** | **0.434** | 0.439 | 0.465 | **0.978** | **0.723** | 1.293 | 0.845 |
| Electricity | 96 | **0.201** | **0.317** | 0.258 | 0.356 | **0.197** | **0.312** | 0.221 | 0.337 | **0.344** | **0.421** | 0.407 | 0.465 |
| | 192 | **0.200** | **0.314** | 0.259 | 0.357 | **0.219** | **0.329** | 0.251 | 0.357 | **0.344** | **0.426** | 0.407 | 0.469 |
| | 336 | **0.214** | **0.330** | 0.268 | 0.364 | **0.263** | **0.359** | 0.288 | 0.380 | **0.358** | **0.440** | 0.392 | 0.461 |
| | 720 | **0.239** | **0.350** | 0.285 | 0.368 | **0.290** | **0.380** | 0.309 | 0.397 | **0.386** | **0.452** | 0.391 | 0.453 |

recorded influenza-like illness patients from the US Center for Disease Control and Prevention. We consider horizon lengths of 24, 32, 48, and 64 with an input length of 32.

**Implementation details**: Following previous work (Xu et al., 2021; Zhou et al., 2021), we pass $\mathbf{X}^{\text{enc}} = \mathbf{X}^{(L)}$ as the input to the encoder. While an array of zero-values would be the default to pass to the decoder, the decoder instead takes as input the second half of the look-back window padded with zeros $\mathbf{X}^{\text{dec}} = \{\mathbf{x}_{t_0+\ell_L/2}, ..., \mathbf{x}_{\ell_L}, 0, 0, ..., 0\}$ with length $\ell_L/2 + \ell_H$. The hidden dimension of models is 512 with a batch size of 32. We use the Adam optimizer with a learning rate of 1e-4. The look-back window size is fixed to 96, and the horizon is varied from 96 to 720. We repeat each experiment 5 times and report average values to reduce randomness. For additional implementation details on our model and baselines please refer to Appendix A.

**Main results and comparison with baselines**: Table 1 shows the results of the proposed framework compared to the baselines. Our proposed multi-scale framework with the adaptive loss outperforms the baselines in almost all of the experiments with an average improvement of 5.6% over FEDFormer, 13% over Autoformer and 38% over Informer which are the three most recent transformer-based architectures on MSE. We also achieved significant error reduction on MAE. The improvement is statistically significant in all cases, and in certain cases quite substantial. In particular for the exchange-rate dataset, with Informer and Reformer base models, our approach improves upon the respective baselines by over 50% averaged over the different horizon lengths.

**Time and memory complexity**: The proposed framework uses the same number of parameters for the model as the baselines (except two parameters $\alpha$ and $c$ of the Adaptive loss). Our framework sacrifices a small amount of computation efficiency for the sake of a significant performance improvement. We expand our analysis in Appendix C. As shown in Table 4 in Appendix, if we replace the operation at the final scale by an interpolation of the prior output, we can achieve improved performance over the baselines, at no computational overhead.

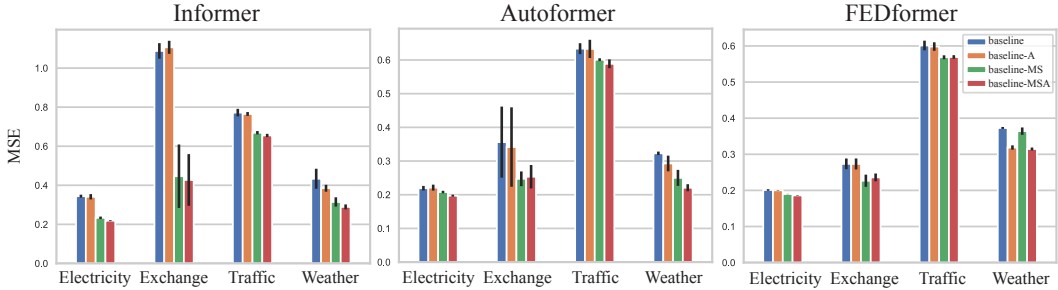

Figure 4: Comparison of training with Adaptive loss "-**A**", multi-scale framework with MSE loss "-**MS**", Multi-scale framework and Adaptive loss "-**MSA**". It shows the combination of all our proposed contributions is essential, and results in significantly improved performance.

Table 3: Single-scale framework with cross scale normalization "**-N**". The cross-scale normalization (which in the single-scale case corresponds to mean-normalization of the output) does not improve the performance of the Autoformer, as it already has an internal trend-cycle normalization component. However, it does improve the results of the Informer and FEDformer.

| Dataset | | FEDformer | | FEDformer-N | | Autoformer | | Autoformer-N | | Informer | | Informer-N | |
|---|---|---|---|---|---|---|---|---|---|---|---|---|---|
| Metric | | MSE | MAE | MSE | MAE | MSE | MAE | MSE | MAE | MSE | MAE | MSE | MAE |
| Weather | 96 | 0.288 | 0.365 | **0.234** | **0.292** | **0.267** | **0.334** | 0.323 | 0.401 | 0.388 | 0.435 | **0.253** | **0.333** |
| | 192 | 0.368 | 0.425 | **0.287** | **0.337** | **0.323** | **0.376** | 0.531 | 0.543 | 0.433 | 0.453 | **0.357** | **0.408** |
| | 336 | 0.447 | 0.469 | **0.436** | **0.443** | **0.364** | **0.397** | 0.859 | 0.708 | 0.610 | 0.551 | **0.459** | **0.461** |
| | 720 | 0.640 | 0.574 | **0.545** | **0.504** | **0.425** | **0.434** | 1.682 | 1.028 | 0.978 | 0.723 | **0.870** | **0.676** |
| Electricity | 96 | 0.201 | 0.317 | **0.194** | **0.307** | **0.197** | **0.312** | 0.251 | 0.364 | 0.344 | 0.421 | **0.247** | **0.356** |
| | 192 | 0.200 | 0.314 | **0.195** | **0.304** | **0.219** | **0.329** | 0.263 | 0.372 | 0.344 | 0.426 | **0.291** | **0.394** |
| | 336 | 0.214 | 0.330 | **0.200** | **0.310** | **0.263** | **0.359** | 0.276 | 0.388 | 0.358 | 0.440 | **0.321** | **0.416** |
| | 720 | 0.239 | 0.350 | **0.225** | **0.332** | 0.290 | **0.380** | **0.280** | 0.385 | 0.386 | 0.452 | **0.362** | **0.434** |

## 4.2 ABLATION STUDY

We present main ablation studies in this section, more ablation results are shown in Appendix G.

**Impact of each component**: Two important components of our approach are the multi-scale framework and the use of the adaptive loss. We conduct multiple experiments (1) removing the multi-scale framework and/or (2) replacing the Adaptive loss by the MSE for training, in order to demonstrate the benefits of these two components. Figure 4 shows the effect of multi-scale and the loss function with different base models. Considering the impact of ablating the adaptive loss, we can see that for both the multi-scale and base models, training with the adaptive loss improves performance. Similarly, adding the multi-scale framework improves performance, both with and without the adaptive loss. Overall, combining the adaptive loss with the multi-scale framework results in the best performance.

**Cross-scale normalization**: As we discussed in Section 3.3, having the cross-scale normalization is crucial to avoid the distribution shifts. To confirm that, we conduct two experiments. Firstly, we use the multi-scale framework without the cross-scale normalization to argue that the error accumulation and covariate shift between the scales leads higher error compared to only a single scale. As shown in Table 2, while the multi-scale framework can get better results in a few cases, it mostly have higher errors than baselines.

On the other hand, adding the normalization with only a single scale can still help to achieve better performance by reducing the effect of covariate shift between the training and the test series. As shown in Table 3, the normalization improves the results of Informer and FEDformer consistently. The decomposition layer of the Autoformer solves a similar problem and replacing that with our normalization harms the capacity of the model.

## 4.3 QUALITATIVE RESULTS

We have also shown the qualitative comparisons between the vanilla Informer and FEDformer versus the results of our framework in Figure 5. Most notably, in both cases our approach appears significantly better at forecasting the statistical properties (such as local variance) of the signal. Our scaleformer-based models capture trends (and other human-relevant) information better than their baselines. Despite these interesting findings, we would like to emphasize that these are randomly selected qualitative examples that may have their own limitations. For more qualitative results, please refer to Section I in the Appendix.

## 4.4 EXTENSIONS AND DISCUSSION

The Scaleformer structural prior has been shown to be beneficial when applied to transformer-based, deterministic time series forecasting. It is not however limited to those settings. In this section, we show it can be extended to *probabilisitc forecasting* and non transformer-based encoders, both of which are closely coupled with our primary application. We also aim to highlight potential promising future directions.

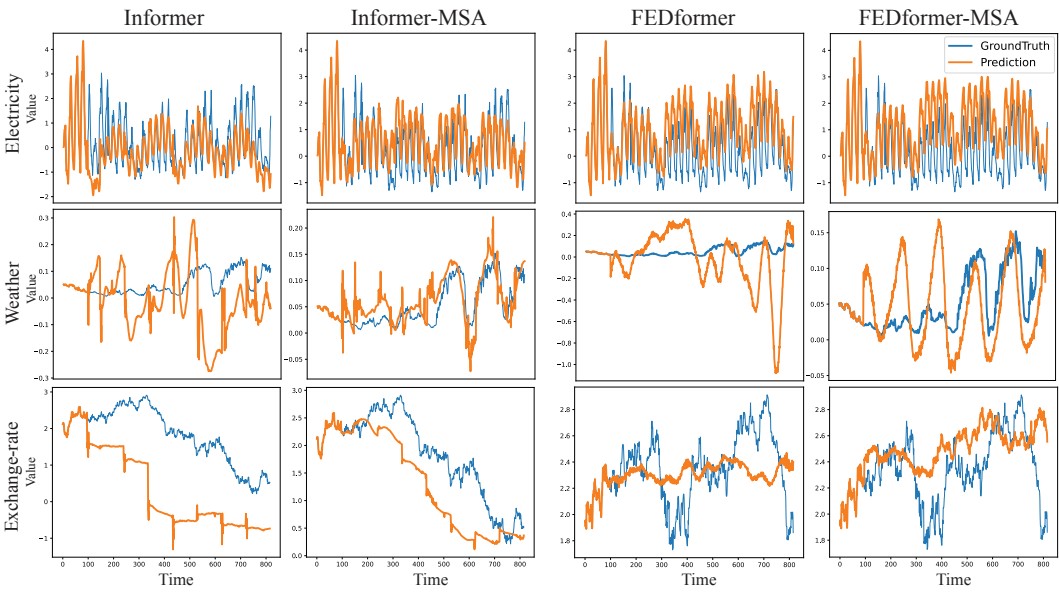

Figure 5: Qualitative comparison of base models with our framework. Our multi-scale models can better capture the global trend and local variations of the signal than their baseline equivalents. To keep the figure concise and readable, we only show Informer and FEDformer, please refer to Figure 8 in Appendix for more results.

We show that our Scaleformer can improve performance in a probabilistic forecasting setting (please refer to Table 9 in Appendix for more details). We adopt the probabilistic output of DeepAR (Salinas et al., 2020), which is the most common probabilistic forecasting treatment. In this setting, instead of a point estimate, we have two prediction heads, predicting the mean $\mu$ and standard deviation $\sigma$, trained with a negative log likelihood loss (NLL). NLL and continuous ranked probability score (CRPS) are used as evaluation metrics. All other hyperparameters remain unchanged. Here, again, scaleformers continue to outperform the probabilistic Informer.

While we have mainly focused on improving transformer-based models, they are not the only encoders. Recent models such as NHits (Challu et al., 2022) and FiLM (Zhou et al., 2022a) attain competitive performance, while assuming a fixed length univariate input/output. They are less flexible compared with variable length of multi-variate input/output, but result in strong performance and faster inference than transformers, making them interesting to consider. The Scaleformer prior demonstrates a statistically significant improvement, on average, when adapted by NHits and FiLM to iteratively refine predictions. For more details please refer to Appendix K.

The results mentioned above demonstrate that ScaleFormer can adapt to settings distinct from point-wise time-series forecasts with transformers (the primary scope of our paper), such as probabilistic forecasts and non-transformer models. We consider such directions to therefore be promising for future work.

## 5 CONCLUSION

Noting that introducing structural priors that account for multi-scale information is essential for accurate time-series forecastings, this paper proposes a novel multi-scale framework on top of recent state-the-art methods for time-series forecasting using Transformers. Our framework iteratively refines a forecasted time-series at increasingly fine-grained scales, and introduces a normalization scheme that minimizes distribution shifts between scales. These contributions result in vastly improved performance over baseline transformer architectures, across a variety of settings, and qualitatively result in forecasts that better capture the trend and local variations of the target signal. Additionally, our detailed ablation study shows that the different components synergetically work together to deliver this outcome.

For future work, it is promising to extend the preliminary work that has been done applying Scale-Former architectures to both probabilistic forecasting and non-transformer models.

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

## A    IMPLEMENTATION DETAILS

Our implementation is based on the Pytorch (Paszke et al., 2019) implementation[5] of the Autoformer and Informer (Xu et al., 2021). The hidden dimension of models are fixed to 512 with a batch size of 32, and we train each model for for 10 epochs with early stop enabled. To optimize the models, an Adam optimizer has been used with a learning rate of 1e-4 for the forecasting model and 1e-3 for optimizing the adaptive loss. The forecasting module is fixed to 2 encoder layers and 1 decoder layer. The look-back window size is fixed to 96, and the horizon is varied from 96 to 720. We repeat each experiment 5 times to reduce the effect of randomness in the reported values. In all experiments, the temporal scale factor $s$ is fixed to 2.

For Informer, we train the model without any changes as the core of our framework. However, Autoformer uses a decomposition layer at the input of the decoder and does not pass the trend series to the network, which makes the model unaware of the previous predictions. To resolve this, we pass zeros as the trend and the series without decomposition as the input to the decoder. For Reformer, we used the available implementation[6] of the model from Xu et al. (2021). In addition, we used the pytorch library[7] of Performer (Choromanski et al., 2020) for our performer baseline using the same parameters as our Reformer model, and finally, we use the official implementation[8] of FEDformer (Zhou et al., 2022b) for the FEDformer model. We fixed the number of modes to 64 following the original paper and Wavelet Enhanced Structure for the core modules. To make it consistent with our other experiments, we use the moving average with the kernel size 25 as in Xu et al. (2021), however, FEDformer (Zhou et al., 2022b) is using moving average with the kernel size of 24. Our random seed is fixed to 2022 in all experiments.

## B    MORE MOTIVATIONS

This section aims to provide more motivations for the use of a multi-scale architecture. Let us first consider the following classical example, highlighted in section 2 of Ferreira et al. (2006), corresponding to the monthly flows of the Fraser River from January of 1913 to December of 1990. As shown in the their corresponding plot, the annual averages are strongly inter-related, pointing to the fact that seasonality alone will not suffice to model the variations. In the context of the paper, this showcases a failure mode of an ARMA model, but this failing is more general: models that do not explicitly account for inter-scale dependencies will perform poorly on similar datasets.

Different approaches have attempted to introduce multi-scale processing (Ferreira et al., 2006; Mozer, 1991) in ways that differ from our own approach. The multi-scale temporal structure for music composition is introduced in (Mozer, 1991). Ferreira et al. (2006) proposed a time series model with rich autocorrelation structures by coupling processes evolving at different levels of resolution through time. However, their base models are constrained to simple statistical models, e.g. Autoregressive models.

To conclude, we note the following: (1) the approaches mentioned above have applied multi-scale modeling with success, and (2) we are the first work to explicitly consider a multi-scale prior by construction for transformers.

## C    REDUCING COMPUTATIONAL COST

To obtain an estimation of the total running time of Scaleformer, the running time of each scale as the terms of a geometric progression based on the scale factor $s$ which results the total time of $\frac{1-s^m}{1-s}$ multiplied by the running time of the baseline method. In this regard, Table 4 shows the running times of different experiments where $s = 2$ with the batch size is 32 on a GeForce GTX 1080 Ti GPU with 64 cores. While our current code is not optimised, still the scaleformer of each method takes roughly twice of the baselines which is consistent with the mentioned formula. Note that this is the smallest scale which means that our method is bounded to twice of the baseline and considering a larger scale

---

[5]https://github.com/thuml/Autoformer

[6]https://github.com/thuml/Autoformer/blob/main/models/Reformer.py

[7]https://github.com/lucidrains/performer-pytorch

[8]https://github.com/MAZiqing/FEDformer

Table 4: Quantitative comparison of training and the test time of different experiments. Using a scale factor 2 the Scaleformer version of the baselines (showed by **-MSA**) should takes roughly twice of the original baseline time which is almost the same in the quantitative comparisons.

| Method | Training time (s) | | | | Test time (s) | | | |
|---|---|---|---|---|---|---|---|---|
| Output Length | 96 | 192 | 336 | 720 | 96 | 192 | 336 | 720 |
| Informer | 11.38 | 13.83 | 17.38 | 25.31 | 0.79 | 1.12 | 1.21 | 1.23 |
| Informer-MSA | 28.73 | 31.88 | 36.37 | 51.15 | 2.42 | 2.63 | 2.87 | 2.96 |
| Autoformer | 17.34 | 23.34 | 31.35 | 52.34 | 1.94 | 2.29 | 2.88 | 3.38 |
| Autoformer-MSA | 40.57 | 51.17 | 63.53 | 111.84 | 5.22 | 6.06 | 7.04 | 8.10 |

factor can reduce the time overhead. Table 5 shows the impact of replacing the Scaleformer operation at the real scale by an interpolation of values of the previous scale, i.e., at scale $s^0$ we do not apply the transformer but rather compute the results by linear interpolation of $\mathbf{X}^{\text{out}}_{m-1}$, thereby reducing the compute cost. This lower cost alternative results in better performance than the baselines, yet worse results than the full Scaleformer. This shows that adding scale is indeed more efficient, and that there is the possibility of a trade-off between further improved performance (full Scaleformer) or improved computational efficiency (interpolated Scaleformer).

Table 5: Comparison of the MSE and MAE results for our proposed multi-scale framework version of Informer and Autoformer by removing the last step and using an interpolation instead (-**MSA**$_r$) with the corresponding original models as the baseline. Results are given in the *multi-variate* setting, for different lenghts of the horizon window. The look-back window size is fixed to 96 for all experiments. The best results are shown in **Bold**. Our method outperforms vanilla versions of both Informer and Autoformer over almost all datasets and settings.

| Dataset | | Autoformer | | Autoformer-**MSA**$_r$ | | Informer | | Informer-**MSA**$_r$ | |
|---|---|---|---|---|---|---|---|---|---|
| Metric | | MSE | MAE | MSE | MAE | MSE | MAE | MSE | MAE |
| Exchange | 96 | 0.154±0.01 | 0.285±0.00 | **0.132**±0.02 | **0.265**±0.02 | 0.966±0.10 | 0.792±0.04 | **0.248**±0.07 | **0.366**±0.06 |
| | 192 | **0.356**±0.09 | **0.428**±0.05 | 0.418±0.22 | 0.466±0.12 | 1.088±0.04 | 0.842±0.01 | **0.727**±0.19 | **0.637**±0.09 |
| | 336 | **0.441**±0.02 | **0.495**±0.01 | 0.736±0.25 | 0.629±0.10 | 1.598±0.08 | 1.016±0.02 | **0.643**±0.03 | **0.620**±0.02 |
| | 720 | 1.118±0.04 | 0.819±0.02 | **0.773**±0.26 | **0.709**±0.13 | 2.679±0.22 | 1.340±0.06 | **1.036**±0.08 | **0.803**±0.03 |
| Weather | 96 | 0.267±0.03 | 0.334±0.02 | **0.168**±0.01 | **0.239**±0.02 | 0.388±0.04 | 0.435±0.03 | **0.211**±0.01 | **0.278**±0.01 |
| | 192 | 0.323±0.01 | 0.376±0.00 | **0.226**±0.01 | **0.296**±0.01 | 0.433±0.05 | 0.453±0.03 | **0.288**±0.03 | **0.336**±0.02 |
| | 336 | 0.364±0.02 | 0.397±0.01 | **0.298**±0.02 | **0.351**±0.02 | 0.610±0.04 | 0.551±0.02 | **0.459**±0.02 | **0.445**±0.02 |
| | 720 | 0.425±0.01 | **0.434**±0.01 | **0.412**±0.04 | **0.434**±0.03 | 0.978±0.05 | 0.723±0.02 | **0.593**±0.07 | **0.528**±0.04 |
| Electricity | 96 | 0.197±0.01 | 0.312±0.01 | **0.189**±0.00 | **0.305**±0.00 | 0.344±0.00 | 0.421±0.00 | **0.194**±0.00 | **0.308**±0.00 |
| | 192 | 0.219±0.01 | 0.329±0.01 | **0.207**±0.00 | **0.323**±0.00 | 0.344±0.01 | 0.426±0.01 | **0.212**±0.00 | **0.327**±0.00 |
| | 336 | 0.263±0.04 | 0.359±0.03 | **0.225**±0.01 | **0.339**±0.00 | 0.358±0.01 | 0.440±0.01 | **0.246**±0.00 | **0.357**±0.00 |
| | 720 | 0.290±0.05 | 0.380±0.02 | **0.250**±0.01 | **0.361**±0.01 | 0.386±0.00 | 0.452±0.00 | **0.283**±0.01 | **0.386**±0.01 |
| Traffic | 96 | 0.628±0.02 | 0.393±0.02 | **0.585**±0.01 | **0.365**±0.01 | 0.748±0.01 | 0.426±0.01 | **0.595**±0.00 | **0.362**±0.00 |
| | 192 | 0.634±0.01 | 0.401±0.01 | **0.606**±0.01 | **0.375**±0.01 | 0.772±0.02 | 0.436±0.01 | **0.629**±0.00 | **0.381**±0.00 |
| | 336 | **0.619**±0.01 | **0.385**±0.01 | 0.631±0.02 | 0.400±0.01 | 0.868±0.04 | 0.493±0.03 | **0.692**±0.01 | **0.410**±0.01 |
| | 720 | **0.656**±0.01 | **0.403**±0.01 | 0.660±0.00 | 0.418±0.01 | 1.074±0.02 | 0.606±0.01 | **0.803**±0.01 | **0.461**±0.00 |

## D JUSTIFICATION FOR THE ADAPTIVE LOSS

Mathematically, the justification for the adaptive loss is as follows. Considering the $\xi$ term in equation 10, the function is asymptotically close (but not equivalent due to the denominator) to $\xi^\alpha$. As a result, for outliers (for which $\xi$ will be large), the loss term will function as a $L^\alpha$ penalty on $\xi$, which will penalize outliers more for large $\alpha$. The converse of this is that we would expect a model trained with such a loss to learn lower values of $\alpha$ for settings with fewer outliers.

## E INTERPLAY BETWEEN ADAPTIVE LOSS AND MULTI-SCALE ARCHITECTURE

The main reason for combining the adaptive loss and multi-scale architecture under a unified framework is: they are synergetic. How do we explain this synergy? Compared to other transformer

Table 6: Comparison of the MSE and MAE results for our proposed multi-scale framework version of Informer and Autoformer (**-MSA**) with the iterative refinemenet baseline of keeping the original scale in each iteration (**-IA**)

| Dataset | | Autoformer-**MSA** | | Autoformer-**IA** | | Informer-**MSA** | | Informer-**IA** | |
|---|---|---|---|---|---|---|---|---|---|
| Metric | | MSE | MAE | MSE | MAE | MSE | MAE | MSE | MAE |
| Exchange | 96 | **0.126**±0.01 | **0.259**±0.01 | 0.410±0.05 | 0.485±0.03 | **0.168**±0.05 | **0.298**±0.03 | 0.649±0.08 | 0.632±0.04 |
| | 192 | **0.253**±0.03 | **0.373**±0.02 | 0.809±0.14 | 0.698±0.06 | **0.427**±0.12 | **0.484**±0.06 | 0.938±0.03 | 0.761±0.01 |
| Weather | 96 | **0.163**±0.01 | **0.226**±0.01 | 0.233±0.01 | 0.293±0.00 | **0.210**±0.02 | **0.279**±0.02 | 0.222±0.01 | 0.286±0.02 |
| | 192 | **0.221**±0.01 | **0.290**±0.02 | 0.401±0.02 | 0.429±0.01 | **0.289**±0.01 | **0.333**±0.01 | 0.357±0.02 | 0.393±0.02 |
| Electricity | 96 | **0.188**±0.00 | **0.303**±0.01 | 0.248±0.01 | 0.360±0.01 | **0.203**±0.01 | **0.315**±0.01 | 0.237±0.00 | 0.344±0.00 |
| | 192 | **0.197**±0.00 | **0.310**±0.00 | 0.265±0.01 | 0.373±0.01 | **0.219**±0.00 | **0.331**±0.00 | 0.271±0.01 | 0.374±0.01 |
| Traffic | 96 | **0.567**±0.00 | **0.350**±0.00 | 0.681±0.03 | 0.432±0.01 | **0.597**±0.01 | 0.369±0.00 | 0.641±0.01 | **0.367**±0.01 |
| | 192 | **0.589**±0.01 | **0.360**±0.01 | 0.699±0.02 | 0.446±0.01 | **0.655**±0.01 | 0.399±0.01 | 0.695±0.01 | **0.388**±0.01 |

architectures (notably the baselines used), ScaleFormer has more iterative steps: the sequential multi-scale operations. Iterative computation tends to accumulate more errors, which will behave like outliers for the purpose of this loss. As a result, the process that leads to the need for the two components can be expressed as: (1) The multi-scale architecture is beneficial for performance as a useful structural prior for time series data. (2) The multi-scale architecture however relies on sequential computation that increases the likelihood of explosive error accumulation. (3) The adaptive loss serves to mitigate this issue, leading to more stable learning and better performance.

## F  SINGLE-SCALE MODEL WITH MEAN-NORMALIZATION

A single-scale model with mean normalization performs better than the multi-scale version without normalization. The reason for this is that normalization as our proposed scheme is targeting at all forms of internal distribution shift, not only those induced by the multi-scale architecture. In our submission, we make the case for the multi-scale prior as a natural prior to add to transformers. From empirical observations, we found that such a prior requires adapting normalization. When investigating means of normalizing, we observed additional benefits to non-multiscale architectures as well. This means that they also suffer from other forms of distribution shift: we attribute it in the paper to e.g. shifts between lookback and forecast distributions.

## G  MORE ABLATION STUDIES AND PARAMETER ANALYSIS

### G.1  ITERATIVE REFINEMENT USING THE SAME SCALE

To further confirm the effect of the multi-scale framework, we also compare our proposed framework with another baseline by keeping the original scale in each iteration. As Table 6 shows, using multi-scale framework outperforms keeping the original scale while having significantly lower memory and time complexity overhead. Following the main experiments, we use the scale factor of 2 which results $S = \{16, 8, 4, 2, 1\}$ for Scaleformer (showed by **-MSA**) and a scale factor of 1 with similarly 5 iterations for iterative refinement (showed by **-IA**).

## G.2 MORE RESULTS ON DIFFERENT COMPONENTS

Table 7 extends the results of Figure 4 of the paper to all four datasets for both Autoformer and Informer, in the multivariate setting and for the two backbones, Autoformer and Informer.

Table 7: Comparison of training with either only an Adaptive loss "-**A**", only the multi-scale framework with MSE loss "-**MS**", or the whole Multi-scale framework and Adaptive loss "-**MSA**". The results confirm that the combination of all our proposed contributions is essential, and results in significantly improved performance in both Informer and Autoformer. Experiments are done in the multi-variate setting.

| Dataset | | Informer | | Informer-**A** | | Informer-**MS** | | Informer-**MSA** | |
|---|---|---|---|---|---|---|---|---|---|
| Metric | | MSE | MAE | MSE | MAE | MSE | MAE | MSE | MAE |
| Exchange | 96 | 0.966±0.10 | 0.792±0.04 | 0.962±0.09 | 0.796±0.04 | 0.201±0.05 | 0.325±0.03 | **0.168**±0.05 | **0.298**±0.03 |
| | 192 | 1.088±0.04 | 0.842±0.01 | 1.106±0.03 | 0.853±0.01 | 0.446±0.14 | 0.502±0.07 | **0.427**±0.12 | **0.484**±0.06 |
| | 336 | 1.598±0.08 | 1.016±0.02 | 1.602±0.07 | 1.019±0.01 | 0.520±0.05 | 0.540±0.02 | **0.500**±0.05 | **0.535**±0.02 |
| | 720 | 2.679±0.22 | 1.340±0.06 | 2.719±0.19 | 1.351±0.06 | **0.997**±0.08 | **0.783**±0.02 | 1.017±0.05 | 0.790±0.02 |
| Weather | 96 | 0.388±0.04 | 0.435±0.03 | 0.342±0.03 | 0.382±0.02 | 0.249±0.02 | 0.324±0.01 | **0.210**±0.02 | **0.279**±0.02 |
| | 192 | 0.433±0.05 | 0.453±0.03 | 0.385±0.02 | 0.404±0.01 | 0.315±0.02 | 0.380±0.02 | **0.289**±0.01 | **0.333**±0.01 |
| | 336 | 0.610±0.04 | 0.551±0.02 | 0.656±0.10 | 0.550±0.04 | 0.473±0.04 | 0.478±0.02 | **0.418**±0.04 | **0.427**±0.03 |
| | 720 | 0.978±0.05 | 0.723±0.02 | 0.953±0.04 | 0.680±0.02 | 0.664±0.04 | 0.585±0.02 | **0.595**±0.04 | **0.532**±0.02 |
| Electricity | 96 | 0.344±0.00 | 0.421±0.00 | 0.321±0.00 | 0.402±0.00 | 0.211±0.00 | 0.326±0.00 | **0.203**±0.01 | **0.315**±0.01 |
| | 192 | 0.344±0.01 | 0.426±0.01 | 0.341±0.01 | 0.422±0.01 | 0.233±0.01 | 0.348±0.00 | **0.219**±0.00 | **0.331**±0.00 |
| | 336 | 0.358±0.01 | 0.440±0.01 | 0.344±0.00 | 0.422±0.00 | 0.279±0.01 | 0.388±0.01 | **0.253**±0.01 | **0.360**±0.01 |
| | 720 | 0.386±0.00 | 0.452±0.00 | 0.363±0.00 | 0.432±0.00 | 0.315±0.01 | 0.411±0.00 | **0.293**±0.01 | **0.390**±0.01 |
| Traffic | 96 | 0.748±0.01 | 0.426±0.01 | 0.744±0.02 | 0.413±0.01 | 0.602±0.01 | 0.375±0.01 | **0.597**±0.01 | **0.369**±0.00 |
| | 192 | 0.772±0.02 | 0.436±0.01 | 0.765±0.01 | 0.416±0.01 | 0.669±0.01 | 0.412±0.00 | **0.655**±0.01 | **0.399**±0.01 |
| | 336 | 0.868±0.04 | 0.493±0.03 | 0.852±0.05 | 0.461±0.03 | 0.815±0.03 | 0.501±0.02 | **0.761**±0.03 | **0.455**±0.03 |
| | 720 | 1.074±0.02 | 0.606±0.01 | 1.030±0.05 | 0.539±0.03 | 0.949±0.01 | 0.556±0.02 | **0.924**±0.02 | **0.521**±0.01 |

| Dataset | | Autoformer | | Autoformer-**A** | | Autoformer-**MS** | | Autoformer-**MSA** | |
|---|---|---|---|---|---|---|---|---|---|
| Metric | | MSE | MAE | MSE | MAE | MSE | MAE | MSE | MAE |
| Exchange | 96 | 0.154±0.01 | 0.285±0.00 | 0.152±0.01 | 0.282±0.00 | 0.145±0.02 | 0.276±0.01 | **0.126**±0.01 | **0.259**±0.01 |
| | 192 | 0.356±0.09 | 0.428±0.05 | 0.342±0.10 | 0.421±0.06 | **0.247**±0.02 | **0.366**±0.01 | 0.253±0.03 | 0.373±0.02 |
| | 336 | **0.441**±0.02 | **0.495**±0.01 | 0.566±0.22 | 0.554±0.10 | 0.456±0.08 | 0.514±0.05 | 0.519±0.16 | 0.538±0.09 |
| | 720 | 1.118±0.04 | 0.819±0.02 | 1.120±0.24 | 0.811±0.10 | **0.729**±0.16 | **0.679**±0.08 | 0.928±0.23 | 0.751±0.09 |
| Weather | 96 | 0.267±0.03 | 0.334±0.02 | 0.229±0.01 | 0.288±0.01 | 0.174±0.01 | 0.254±0.01 | **0.163**±0.01 | **0.226**±0.01 |
| | 192 | 0.323±0.01 | 0.376±0.00 | 0.293±0.02 | 0.340±0.02 | 0.250±0.02 | 0.333±0.02 | **0.221**±0.01 | **0.290**±0.02 |
| | 336 | 0.364±0.02 | 0.397±0.01 | 0.357±0.01 | 0.387±0.01 | 0.314±0.02 | 0.380±0.02 | **0.282**±0.02 | **0.340**±0.03 |
| | 720 | 0.425±0.01 | 0.434±0.01 | 0.419±0.01 | 0.422±0.01 | 0.414±0.03 | 0.457±0.02 | **0.369**±0.04 | **0.396**±0.03 |
| Electricity | 96 | 0.197±0.01 | 0.312±0.01 | 0.201±0.01 | 0.312±0.01 | 0.196±0.00 | 0.312±0.01 | **0.188**±0.00 | **0.303**±0.01 |
| | 192 | 0.219±0.01 | 0.329±0.01 | 0.221±0.01 | 0.328±0.01 | 0.208±0.00 | 0.323±0.00 | **0.197**±0.00 | **0.310**±0.00 |
| | 336 | 0.263±0.04 | 0.359±0.03 | 0.232±0.01 | 0.339±0.01 | **0.220**±0.00 | 0.336±0.00 | 0.224±0.02 | **0.333**±0.01 |
| | 720 | 0.290±0.05 | 0.380±0.02 | **0.249**±0.01 | **0.351**±0.01 | 0.252±0.00 | 0.364±0.00 | 0.249±0.01 | 0.358±0.01 |
| Traffic | 96 | 0.628±0.02 | 0.393±0.02 | 0.610±0.02 | 0.381±0.01 | 0.580±0.01 | 0.358±0.00 | **0.567**±0.00 | **0.350**±0.00 |
| | 192 | 0.634±0.01 | 0.401±0.01 | 0.633±0.02 | 0.396±0.02 | 0.601±0.00 | 0.369±0.00 | **0.589**±0.01 | **0.360**±0.01 |
| | 336 | 0.619±0.01 | 0.385±0.01 | **0.618**±0.00 | **0.381**±0.00 | 0.639±0.02 | 0.397±0.01 | 0.619±0.01 | 0.383±0.01 |
| | 720 | 0.656±0.01 | 0.403±0.01 | 0.654±0.02 | **0.397**±0.01 | 0.680±0.02 | 0.427±0.01 | **0.642**±0.01 | 0.397±0.01 |

## G.3 RESULTS ON DIFFERENT SCALES

Table 8 showcases the results of our model using different values of the scale parameter $s$. It shows that a scale of 2 results in better performance.

Table 8: Comparison of the baselines with our method using different scales. Reducing the scale factor from $s = 16$ to $s = 2$ increases the number of steps but achieves lower error on average.

| Dataset | | Informer | | Informer-MSA(s=16) | | Informer-MSA(s=4) | | Informer-MSA(s=2) | |
|---|---|---|---|---|---|---|---|---|---|
| Metric | | MSE | MAE | MSE | MAE | MSE | MAE | MSE | MAE |
| Exchange | 96 | 0.966±0.10 | 0.792±0.04 | 0.376±0.08 | 0.480±0.04 | 0.246±0.06 | 0.374±0.04 | **0.168**±0.05 | **0.298**±0.03 |
| | 192 | 1.088±0.04 | 0.842±0.01 | 0.878±0.08 | 0.721±0.03 | 0.661±0.20 | 0.617±0.09 | **0.427**±0.12 | **0.484**±0.06 |
| | 336 | 1.598±0.08 | 1.016±0.02 | 0.899±0.07 | 0.733±0.03 | 0.697±0.10 | 0.648±0.05 | **0.500**±0.05 | **0.535**±0.02 |
| | 720 | 2.679±0.22 | 1.340±0.06 | 1.633±0.14 | 1.031±0.05 | 1.457±0.27 | 0.958±0.09 | **1.017**±0.05 | **0.790**±0.02 |
| Weather | 96 | 0.388±0.04 | 0.435±0.03 | 0.208±0.02 | 0.275±0.02 | **0.189**±0.01 | **0.259**±0.01 | 0.210±0.02 | 0.279±0.02 |
| | 192 | 0.433±0.05 | 0.453±0.03 | 0.302±0.02 | 0.354±0.02 | **0.287**±0.02 | **0.333**±0.01 | 0.289±0.01 | 0.333±0.01 |
| | 336 | 0.610±0.04 | 0.551±0.02 | 0.470±0.06 | 0.456±0.04 | 0.420±0.03 | 0.433±0.02 | **0.418**±0.04 | **0.427**±0.03 |
| | 720 | 0.978±0.05 | 0.723±0.02 | 0.639±0.07 | 0.544±0.04 | 0.627±0.07 | 0.543±0.04 | **0.595**±0.04 | **0.532**±0.02 |
| Electricity | 96 | 0.344±0.00 | 0.421±0.00 | 0.215±0.00 | 0.329±0.00 | **0.203**±0.00 | **0.315**±0.00 | **0.203**±0.01 | **0.315**±0.01 |
| | 192 | 0.344±0.01 | 0.426±0.01 | 0.257±0.01 | 0.370±0.01 | 0.235±0.01 | 0.347±0.01 | **0.219**±0.00 | **0.331**±0.00 |
| | 336 | 0.358±0.01 | 0.440±0.01 | 0.300±0.04 | 0.400±0.03 | 0.264±0.01 | 0.373±0.01 | **0.253**±0.01 | **0.360**±0.01 |
| | 720 | 0.386±0.00 | 0.452±0.00 | 0.334±0.03 | 0.418±0.02 | 0.306±0.01 | 0.401±0.01 | **0.293**±0.01 | **0.390**±0.01 |
| Traffic | 96 | 0.748±0.01 | 0.426±0.01 | 0.648±0.02 | 0.386±0.01 | 0.616±0.00 | 0.374±0.01 | **0.597**±0.01 | **0.369**±0.00 |
| | 192 | 0.772±0.02 | 0.436±0.01 | 0.679±0.01 | **0.391**±0.01 | 0.686±0.01 | 0.404±0.01 | **0.655**±0.01 | 0.399±0.01 |
| | 336 | 0.868±0.04 | 0.493±0.03 | 0.811±0.02 | 0.455±0.01 | 0.782±0.03 | **0.451**±0.02 | **0.761**±0.03 | 0.455±0.03 |
| | 720 | 1.074±0.02 | 0.606±0.01 | 1.020±0.07 | 0.542±0.03 | 0.965±0.03 | **0.521**±0.01 | **0.924**±0.02 | **0.521**±0.01 |

| Dataset | | Autoformer | | Autoformer-MSA(16) | | Autoformer-MSA(4) | | Autoformer-MSA(2) | |
|---|---|---|---|---|---|---|---|---|---|
| Metric | | MSE | MAE | MSE | MAE | MSE | MAE | MSE | MAE |
| Exchange | 96 | 0.154±0.01 | 0.285±0.00 | 0.182±0.03 | 0.316±0.03 | 0.170±0.03 | 0.304±0.02 | **0.126**±0.01 | **0.259**±0.01 |
| | 192 | 0.356±0.09 | 0.428±0.05 | 0.514±0.18 | 0.537±0.10 | 0.359±0.14 | 0.443±0.08 | **0.253**±0.03 | **0.373**±0.02 |
| | 336 | **0.441**±0.02 | **0.495**±0.01 | 0.527±0.07 | 0.570±0.04 | 0.606±0.18 | 0.585±0.10 | 0.519±0.16 | 0.538±0.09 |
| | 720 | 1.118±0.04 | 0.819±0.02 | 1.019±0.18 | 0.819±0.08 | 0.973±0.22 | 0.809±0.09 | **0.928**±0.23 | **0.751**±0.09 |
| Weather | 96 | 0.267±0.03 | 0.334±0.02 | 0.169±0.00 | 0.239±0.01 | 0.164±0.00 | 0.234±0.01 | **0.163**±0.01 | **0.226**±0.01 |
| | 192 | 0.323±0.01 | 0.376±0.00 | 0.240±0.03 | 0.310±0.03 | 0.229±0.01 | 0.299±0.02 | **0.221**±0.01 | **0.290**±0.02 |
| | 336 | 0.364±0.02 | 0.397±0.01 | 0.304±0.03 | 0.354±0.03 | 0.303±0.01 | 0.350±0.01 | **0.282**±0.02 | **0.340**±0.03 |
| | 720 | 0.425±0.01 | 0.434±0.01 | 0.375±0.01 | 0.403±0.01 | 0.382±0.02 | 0.414±0.01 | **0.369**±0.04 | **0.396**±0.03 |
| Electricity | 96 | 0.197±0.01 | 0.312±0.01 | 0.190±0.00 | 0.306±0.00 | **0.188**±0.00 | **0.303**±0.00 | **0.188**±0.00 | **0.303**±0.01 |
| | 192 | 0.219±0.01 | 0.329±0.01 | 0.206±0.01 | 0.320±0.01 | 0.207±0.00 | 0.320±0.00 | **0.197**±0.00 | **0.310**±0.00 |
| | 336 | 0.263±0.04 | 0.359±0.03 | 0.236±0.02 | 0.344±0.01 | 0.237±0.03 | 0.344±0.02 | **0.224**±0.02 | **0.333**±0.01 |
| | 720 | 0.290±0.05 | 0.380±0.02 | 0.260±0.01 | 0.368±0.01 | 0.261±0.01 | 0.369±0.01 | **0.249**±0.01 | **0.358**±0.01 |
| Traffic | 96 | 0.628±0.02 | 0.393±0.02 | 0.605±0.01 | 0.380±0.01 | 0.594±0.02 | 0.367±0.01 | **0.567**±0.00 | **0.350**±0.00 |
| | 192 | 0.634±0.01 | 0.401±0.01 | 0.626±0.01 | 0.393±0.01 | 0.600±0.01 | 0.369±0.00 | **0.589**±0.01 | **0.360**±0.01 |
| | 336 | **0.619**±0.01 | 0.385±0.01 | 0.635±0.01 | 0.400±0.00 | 0.625±0.01 | 0.386±0.01 | 0.619±0.01 | **0.383**±0.01 |
| | 720 | 0.656±0.01 | 0.403±0.01 | 0.697±0.03 | 0.439±0.01 | 0.678±0.02 | 0.422±0.02 | **0.642**±0.01 | **0.397**±0.01 |

## G.4 EFFECT OF $\alpha$ ON THE LOSS FUNCTION

Figure 6 shows the impact of $\alpha$ on the shape of the loss function on the left, and example ground truth time-series corresponding to the horizon window on the right. As noted in the main paper, lower values of $\alpha$ tend to result in better robustness to outliers. This is indeed confirmed empirically in the case of the weather dataset, which corresponds to the lowest learnt $\alpha$ value, and has the outliers with the largest relative scales. For simplicity, we have excluded $c$ on the analysis as it does not impact the robustness with regards to the outliers, as shown in Barron (2019).

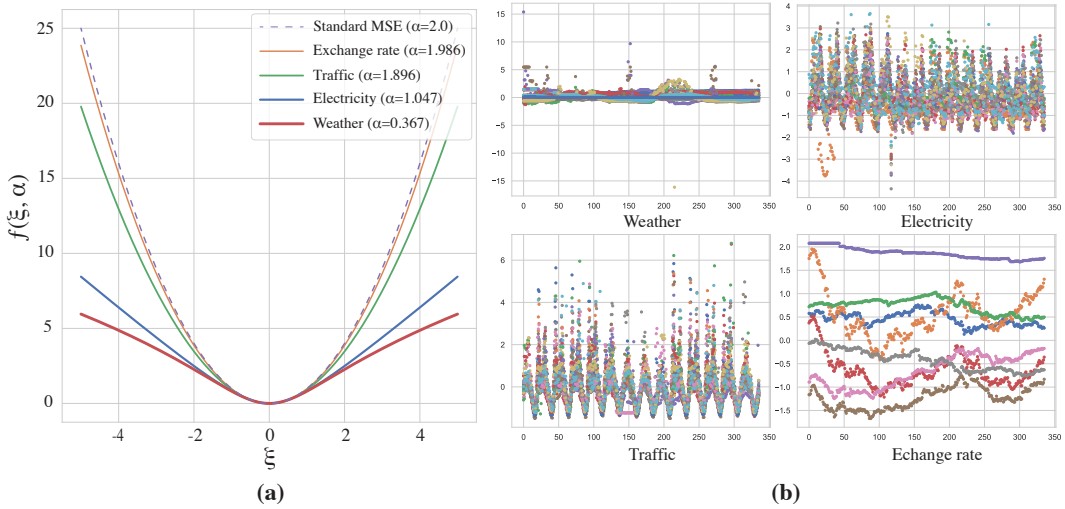

Figure 6: (**a**) The loss value based as a function of the $\xi$ (absolute difference between prediction and target) and the learned $\alpha$ for each dataset. Different colors correspond to different datasets (each with specific value of $\alpha$). (**b**) Samples taken from the ground-truth horizon window of each dataset. The Weather dataset sample has the outliers with the largest scales compared to the input series. As a consequence, the learnt value of $\alpha$ is the lowest. Different colors correspond to different variables in the multi-variate time-series we are considering.

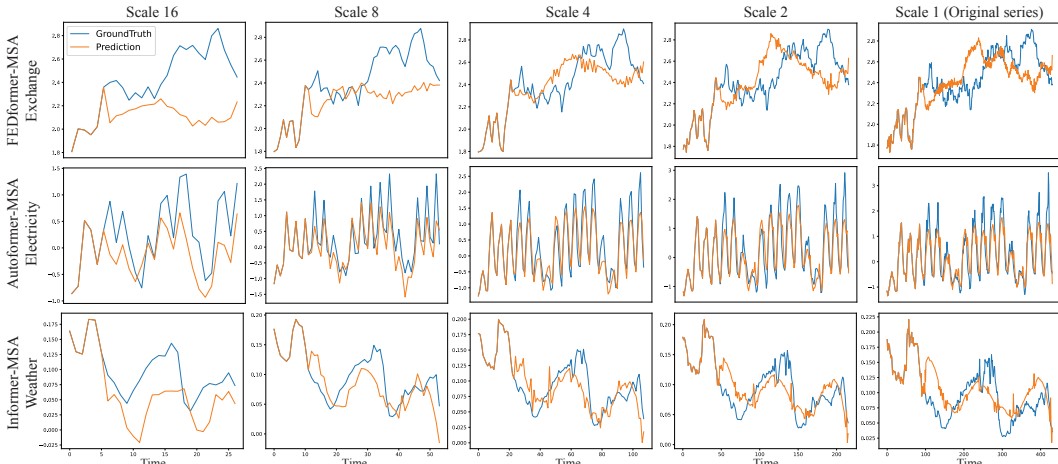

Figure 7: Qualitative results of each scale. The model can correct its previous mistakes on higher scales, showcasing increased robustness to forecasting artifacts that occur individually at each scale.

## H    DISCUSSION ON NORMALIZATION FOR AUTOFORMER

There is limited performance gain for Autoformer from the normalization alone. The reason is that Autoformer benefits from an inner series decomposition module which acts as the normalization by nature. Indeed one benefit of our proposed framework is a simple solution to bring the benefits of these specified designs to other baselines which significantly reduces the gap between for example Informer and Autoformer. AutoFormer already benefits from an internal component that reduces internal distribution shift: the series decomposition module.

## I    ADDITIONAL QUALITATIVE RESULTS

Figure 7, Figure 8, and Figure 9 provide additional qualitative results respectively showing the intermediate results of each scale and the comparison between the baselines and our approach with the horizon length of 720 and 192. As we can see, Scaleformer is able to better learn local and global time-series variations.

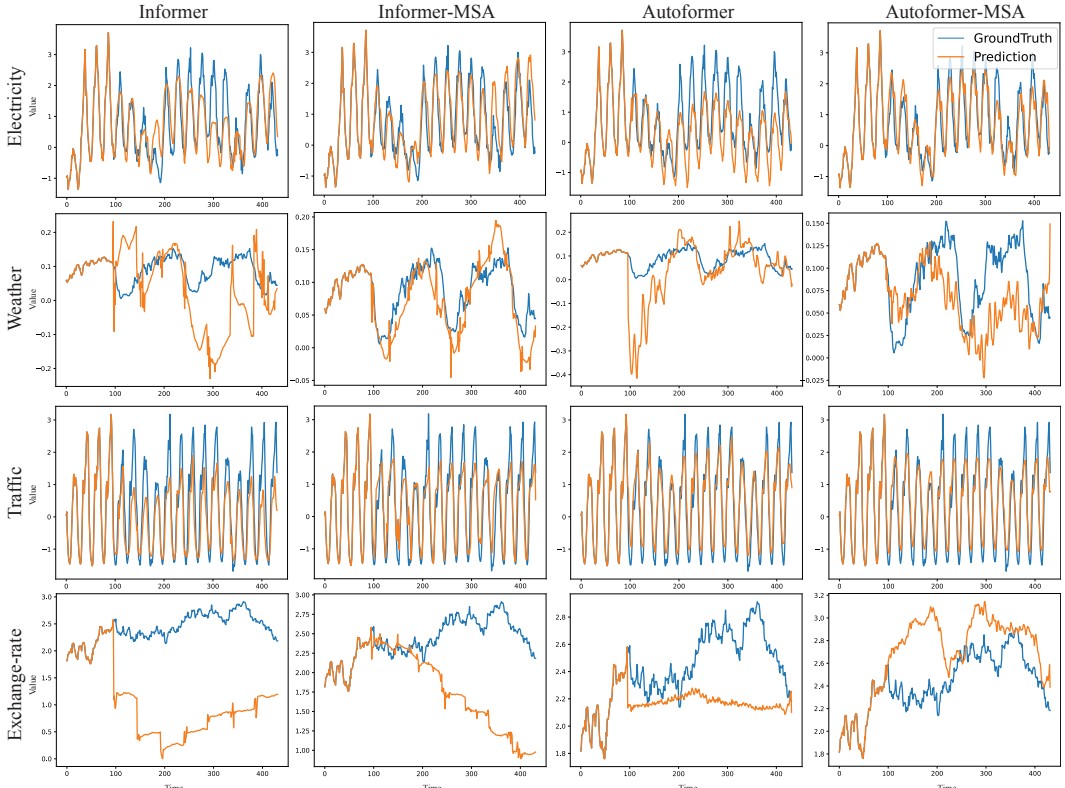

Figure 8: Additional qualitative comparison of the baselines against our framework.

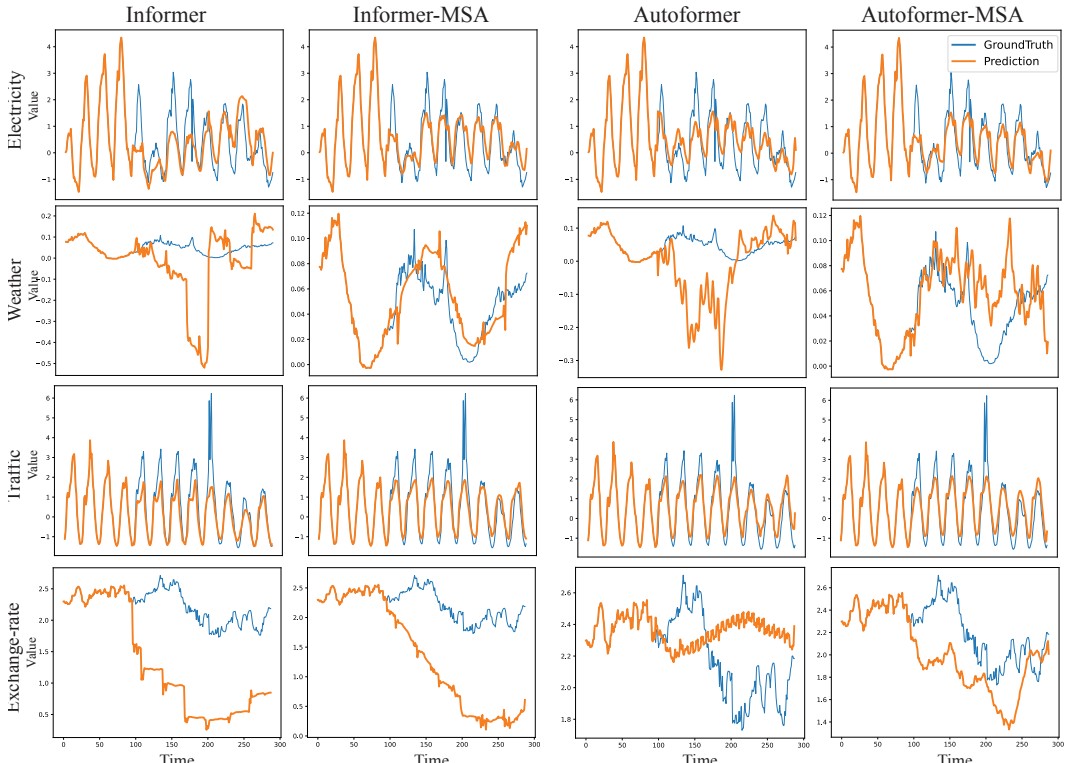

Figure 9: Additional qualitative comparison of the baselines against our framework.

## J    PROBABILISTIC FORECASTING

Table 9 shows the comparison of probabilistic methods for Informer by following the probabilistic output of DeepAR (Salinas et al., 2020), which is the most common probabilistic forecasting treatment. On implementation details, instead of point estimate, we have two prediction heads (one predict $\mu$ and one predict $\sigma$) with negative log likelihood loss. The other hyper parameters is the same as before. We use NLL loss and continuous ranked probability score (CRPS) which are commonly used in probabilistic forecasting as evaluation metrics.

Table 9: Scaleformer improving probabilistic metrics in Probabilistic forecasting for Informer. By adapting Informer to make probabilistic predictions, we are able to show the Scaleformer prior again brings benefits. While such an analysis excludes dedicated probabilistic approaches for conciseness, it nevertheless shows the *generality* of our proposed approach.

| Dataset | | 96 | | 192 | | 336 | | 720 | |
|---------|--------------|------------------|------------------|------------------|------------------|------------------|------------------|------------------|-------------------|
| Metric | | CRPS | NLL | CRPS | NLL | CRPS | NLL | CRPS | NLL |
| Exchange | Informer | 0.548±0.02 | 2.360±0.20 | 0.702±0.05 | 4.350±1.45 | 0.826±0.02 | 4.302±0.49 | 1.268±0.06 | 13.140±1.84 |
| | Informer-MSA | **0.202**±0.01 | **0.452**±0.09 | **0.284**±0.02 | **0.818**±0.11 | **0.414**±0.06 | **1.724**±0.43 | **0.570**±0.03 | **2.862**±0.21 |
| Weather | Informer | 0.376±0.03 | 1.180±0.21 | 0.502±0.03 | 1.752±0.23 | 0.564±0.02 | 1.928±0.27 | 0.684±0.09 | 2.210±0.46 |
| | Informer-MSA | **0.250**±0.02 | **0.392**±0.14 | **0.294**±0.01 | **0.610**±0.04 | **0.308**±0.02 | **0.728**±0.10 | **0.438**±0.04 | **1.270**±0.14 |
| Electricity | Informer | 0.330±0.01 | 1.106±0.05 | 0.338±0.05 | 1.254±0.04 | 0.348±0.01 | 1.244±0.07 | 0.528±0.00 | 1.856±0.06 |
| | Informer-MSA | **0.238**±0.01 | **0.578**±0.01 | **0.290**±0.00 | **0.776**±0.01 | **0.324**±0.03 | **0.904**±0.10 | **0.358**±0.01 | **1.022**±0.04 |
| Traffic | Informer | 0.372±0.04 | 1.376±0.05 | 0.340±0.01 | 1.404±0.04 | 0.372±0.01 | 1.516±0.06 | 0.568±0.01 | 1.658±0.01 |
| | Informer-MSA | **0.288**±0.01 | **1.094**±0.03 | **0.312**±0.01 | **1.102**±0.04 | **0.368**±0.02 | **1.194**±0.05 | **0.442**±0.02 | **1.378**±0.06 |

## K    EXTENSION TO OTHER METHODS

Table 10 shows the comparison results of NHiTs (Challu et al., 2022) and FiLM (Zhou et al., 2022a) as two baselines. For each method, we copy original model to have model for different scales and we concatenate the input with the output of previous scale for the new scale. The training hyperparameters such as optimizer and learning rate is the same as the previous baselines.

Table 10: The effect of applying our proposed framework to NHits and FiLM as two non-transformer based models. Best results are shown in **Bold**.

| Dataset | | NHiTS | | NHiTS-**MSA** | | FiLM | | FiLM-**MSA** | |
|---|---|---|---|---|---|---|---|---|---|
| Metric | | MSE | MAE | MSE | MAE | MSE | MAE | MSE | MAE |
| Exchange | 96 | 0.091±0.00 | 0.218±0.01 | **0.087**±0.00 | **0.206**±0.00 | 0.083±0.00 | 0.201±0.00 | **0.081**±0.00 | **0.197**±0.00 |
| | 192 | 0.200±0.02 | 0.332±0.01 | **0.186**±0.01 | **0.306**±0.00 | 0.179±0.00 | 0.301±0.00 | **0.156**±0.00 | **0.284**±0.00 |
| | 336 | **0.347**±0.03 | **0.442**±0.02 | 0.381±0.01 | 0.445±0.01 | 0.341±0.00 | 0.421±0.00 | **0.253**±0.01 | **0.378**±0.01 |
| | 720 | **0.761**±0.20 | **0.662**±0.08 | 1.124±0.07 | 0.808±0.03 | 0.896±0.01 | 0.714±0.00 | **0.728**±0.01 | **0.659**±0.00 |
| Weather | 96 | 0.169±0.00 | 0.228±0.00 | **0.167**±0.00 | **0.211**±0.00 | **0.194**±0.00 | 0.235±0.00 | 0.195±0.00 | **0.232**±0.00 |
| | 192 | 0.210±0.00 | 0.268±0.00 | **0.208**±0.00 | **0.253**±0.00 | 0.238±0.00 | 0.270±0.00 | **0.235**±0.00 | **0.269**±0.00 |
| | 336 | **0.261**±0.00 | 0.313±0.00 | **0.261**±0.00 | **0.294**±0.00 | 0.288±0.00 | 0.305±0.00 | **0.275**±0.00 | **0.303**±0.00 |
| | 720 | 0.333±0.01 | 0.372±0.01 | **0.331**±0.00 | **0.348**±0.00 | 0.359±0.00 | **0.350**±0.00 | **0.337**±0.00 | 0.356±0.00 |

## L    CROSS-SCALE NORMALIZATION

The choice of normalization approach is essential to our method, due to the aforementioned distribution shifts. In this section, we showcase multiple experiments that were done to study the impact of different ways of normalizing inputs. In Table 11, we compare using two forms of normalization with Informer-MSA as the baseline: mean-only versus normalization using both mean and standard deviation. We consider that adding normalization to our method is a trade-off between two issues faced during training. One one hand, (internal) distribution shifts hinder performance, as demonstrated in the experiments (See Table 2). On the other hand, normalizing the internal representations results in a form of information loss (during the processing of the input): we lose information about mean and standard deviation of the input data. We believe that the reason mean-normalization outperforms mean and standard deviation normalization is because it results in a better trade-off, losing less information while still managing to address the internal distribution shift.

Fig. 10 shows the different dataset distributions. As we can see, mean normalization works better for datasets such as electricity and traffic with almost unimodal distributions. For datasets such as exchange-rate and to some extent weather, which have multi-modal distributions, we find that mean and standandard deviation works better. Our hypothesis is that mean-normalization is too simple an approach for more complex input distributions. We note that, in all cases, **either form of normalization results in strong improvements compared to the absence of normalization**.

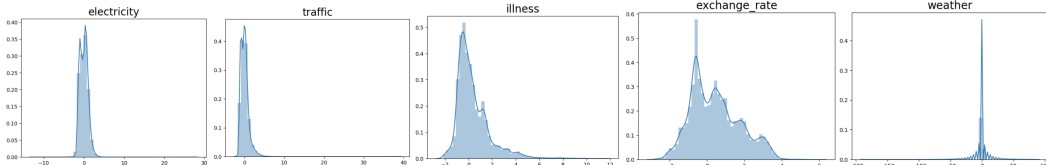

Figure 10: Dataset distribution with histogram and kernel density estimation. Note that different datasets have either mostly uni-modal or multi-modal distributions.

## M    SYNTHETIC DATASET

To further evaluate our proposed framework and adaptive loss function, we generate a new dataset based on the Mackey–Glass equations (Mackey & Glass, 1977). Concretely, we use the following

Table 11: Comparison of using standard normalization and zero-mean shifting in our cross scale normalization. Zero-mean shifting gets better results in almost all of the Traffic, Electricity, and Illness dataset. While standard normalization gets better results in Weather and Exchange datasets. We use zero-mean shifting in our experiments as it removes less information from the input time-series.

| Prediction Length | | 96 (24) | | 192 (32) | | 336 (48) | | 720 (64) | |
|---|---|---|---|---|---|---|---|---|---|
| Metric | | MSE | MAE | MSE | MAE | MSE | MAE | MSE | MAE |
| Electricity | Mean | **0.199**±0.00 | **0.312**±0.00 | **0.215**±0.00 | **0.327**±0.00 | **0.250**±0.01 | **0.358**±0.01 | **0.297**±0.01 | **0.391**±0.01 |
| | Mean+STDEV | 0.248±0.01 | 0.343±0.01 | 0.271±0.02 | 0.363±0.02 | 0.265±0.01 | 0.359±0.00 | 0.365±0.01 | 0.425±0.01 |
| Exchange | Mean | 0.191±0.02 | 0.326±0.02 | 0.374±0.05 | 0.453±0.02 | 0.555±0.03 | 0.567±0.01 | **1.011**±0.06 | **0.782**±0.02 |
| | Mean+STDEV | **0.165**±0.01 | **0.300**±0.01 | **0.280**±0.02 | **0.387**±0.02 | **0.487**±0.13 | **0.518**±0.06 | 1.406±0.20 | 0.931±0.07 |
| ILI | Mean | **3.695**±0.07 | **1.242**±0.01 | **3.798**±0.07 | **1.266**±0.01 | 3.759±0.09 | 1.238±0.01 | **3.606**±0.02 | **1.222**±0.00 |
| | Mean+STDEV | 4.582±0.29 | 1.435±0.07 | 4.681±0.15 | 1.470±0.03 | **3.611**±0.39 | **1.243**±0.07 | 4.156±0.17 | 1.376±0.02 |
| Traffic | Mean | **0.611**±0.01 | **0.377**±0.01 | **0.642**±0.01 | **0.389**±0.01 | **0.760**±0.01 | **0.453**±0.01 | **0.940**±0.00 | **0.514**±0.00 |
| | Mean+STDEV | 0.712±0.02 | 0.421±0.01 | 0.777±0.01 | 0.474±0.01 | 0.852±0.01 | 0.501±0.01 | 0.981±0.02 | 0.530±0.01 |
| Weather | Mean | 0.208±0.01 | 0.274±0.01 | 0.303±0.02 | 0.347±0.02 | 0.443±0.03 | 0.435±0.02 | 0.625±0.04 | 0.544±0.02 |
| | Mean+STDEV | **0.197**±0.01 | **0.242**±0.00 | **0.224**±0.01 | **0.276**±0.01 | **0.301**±0.01 | **0.331**±0.01 | **0.392**±0.02 | **0.384**±0.01 |

equation:

$$\frac{dx}{dt} = \frac{0.2 \times x(t-\tau)}{1 + x(t-\tau)^{10}} - (0.1 \times x(t)), \tag{11}$$

where we follow López-Caraballo et al. (2016); Farmer et al. (1983) and we also consider $x(0) = 1.2$. We create three series of length 10k. First we create a series by only using the above equation and considering $\tau = 18$ where the series has a chaotic behaviour when $\tau \geq 17$. We also provide two additional series with $\tau = 12$ and $\tau = 9$, and also with added seasonal and trend components. Our synthetic dataset is a combination of these 3 series (See Figure 11 for an example). We compare our method using the new dataset on both Informer and Autoformer in Table 12, showing our proposed framework significantly improves the performance over the baselines.

Table 12: Comparison of our proposed framework and the baselines using a synthetic data with 3 series generated based on Mackey Glass formulation by $\tau = \{18, 12, 9\}$.

| Dataset | 96 | | 192 | | 336 | | 720 | |
|---|---|---|---|---|---|---|---|---|
| Metric | MSE | MAE | MSE | MAE | MSE | MAE | MSE | MAE |
| Autoformer_base | 0.435±0.05 | 0.464±0.03 | 0.399±0.06 | 0.456±0.03 | 0.361±0.03 | 0.449±0.02 | **0.361**±0.02 | **0.452**±0.01 |
| Autoformer_MS | **0.075**±0.01 | **0.192**±0.02 | **0.113**±0.02 | **0.237**±0.02 | **0.164**±0.03 | **0.298**±0.03 | 0.468±0.22 | 0.501±0.14 |
| Informer_base | 0.265±0.06 | 0.367±0.03 | 0.287±0.01 | 0.396±0.00 | 0.243±0.02 | 0.372±0.01 | 0.246±0.01 | 0.375±0.01 |
| Informer_MS | **0.078**±0.00 | **0.192**±0.00 | **0.109**±0.01 | **0.233**±0.02 | **0.164**±0.01 | **0.309**±0.01 | **0.227**±0.03 | **0.356**±0.02 |

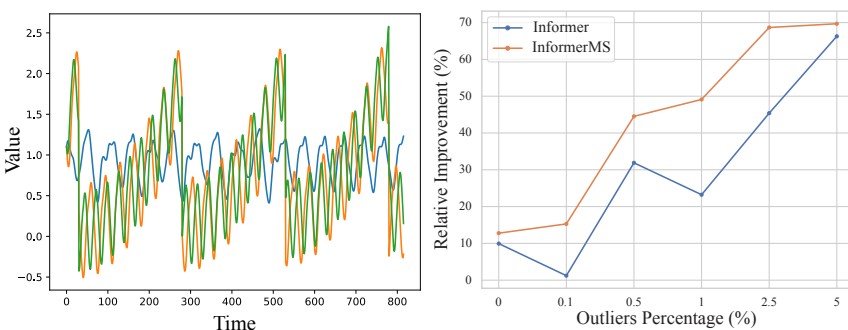

Figure 11: **Left:** an example input of the synthetic dataset. **Right:** The relative improvement of adaptive loss increases with increasing the percentage of outliers in the dataset.

In addition, to further analyze the effect of the adaptive loss on noisy datasets with outliers, we randomly replace a percentage of the training data with outliers by defining an outlier as a sample with a distance of larger than 50 times of standard deviation of the series from the median. Figure 11 shows the improvement of using adaptive loss on both Informer baseline and our multi-scale version of Informer. Adaptive loss increases the performance up to roughly $70\%$ when there is a noisy input with extreme outliers ($5\%$ of data) while it gets comparable results for a clean dataset, demonstrating it as a strong candidate to replace MSE loss in the time series forecasting tasks.

## N  STATISTICAL TESTS

To further validate the strengths of our empirical results, we have conducted two statistical tests. Each test is a Student's t-test Kendall (1960) to determine whether the two sets of results can be distinguished. The first test, in Table 13, shows that for most settings, the adaptive loss provides gains that are statistically significant ($p < 0.05$). The second test, in Table 14 shows that for most settings, the multi-scale architectures provides gains that are statistically significant ($p < 0.5$).

Table 13: Student's t-test p-values for the test corresponding to whether the Adaptive loss provides an improvement. The base model is FedFormer. While we cannot conclude the adaptive loss provides an improvement for exchange rate and certain window sizes for traffic, for all other datasets the improvement is notable.

| Window size | 96 | | 192 | | 336 | | 720 | |
|---|---|---|---|---|---|---|---|---|
| Metric | MSE | MAE | MSE | MAE | MSE | MAE | MSE | MAE |
| Exchange rate | 0.7157 | 0.77286 | 0.46741 | 0.57392 | 0.61317 | 0.68405 | 0.97102 | 0.56755 |
| Electricity | 0.05844 | 0.01267 | 0.00124 | 0.00187 | 0.01294 | 0.00576 | 0.00374 | 0.00597 |
| Weather | 0.25509 | 0.11246 | 0.00014 | 8e-05 | 0.0648 | 0.00826 | 0.05881 | 0.02213 |
| Traffic | 0.21029 | 0.02572 | 0.78931 | 0.19641 | 0.40819 | 0.248 | 0.3491 | 0.17579 |

Table 14: Student's t-test p-values for the test corresponding to whether the multi-scale prior provides an improvement. The base model is FedFormer. For certain window sizes of weather and exchange rate the improvement is not statistically provable. For most other settings it is almost always significant.

| Window size | 96 | | 192 | | 336 | | 720 | |
|---|---|---|---|---|---|---|---|---|
| Metric | MSE | MAE | MSE | MAE | MSE | MAE | MSE | MAE |
| Exchange rate | 0.01109 | 0.01657 | 0.02263 | 0.01674 | 0.14988 | 0.26543 | 0.11071 | 0.13141 |
| Electricity | 0.00029 | 0.00088 | 2e-05 | 0.00017 | 0.28221 | 0.3316 | 0.00593 | 0.07219 |
| Weather | 0.02525 | 0.01987 | 0.2201 | 0.05433 | 0.14593 | 0.12861 | 0.00362 | 0.02396 |
| Traffic | 0.01024 | 0.00241 | 0.01714 | 0.00749 | 0.00058 | 0.00033 | 0.00019 | 0.00012 |

