# OpenReview forum: "Scaleformer: Iterative Multi-scale Refining Transformers for Time Series Forecasting"
_ICLR.cc/2023/Conference — ICLR 2023 poster_

### Official Review · Reviewer_V4CM · 2022-10-25

**Confidence:** 3
**Clarity, Quality, Novelty And Reproducibility:** 1. My major concern is the similarity…
**Correctness:** 4
**Technical Novelty And Significance:** 3
**Empirical Novelty And Significance:** 3
**Recommendation:** 6

**Strength And Weaknesses:**

Strength:
1. the paper is well written and the technique is simple yet intuitive
2. the contribution can contribute any transformer based forecasting model
3. Extensive experiments are conducted to justify the claim

Weakness
1. Introducing Adaptive loss seems unrelated to the major contribution of this paper
2. The multiscale forecasting is so-what similar to U-Net structure

**Summary Of The Paper:**

This paper introduced a multi-scale forecasting training module to improve the forecasting performance of existing transformer models. The basic idea is laying on training scaling-shift aware fine-grind forecasting model. To achieve this goal, the authors introduce an across scale normalization module and a multi-resolution forecasting training structure. The result shows that the proposed module could improve the forecasting performance with various transformer deep learning models without adding additional burdens.


**Summary Of The Review:**

Overall, this is an interesting paper, and the proposed module can be widely used in existing time series forecasting framework. Therefore, I believe this is an accept paper.

---

> ### Author Response · Authors · 2022-11-16
> **Response to reviewer V4CM**
>
> We thank the reviewer for your positive, insightful and valuable comments and suggestions which are very crucial for improving the quality of our manuscript.
>
> 1. Relation of the adaptive loss to the main paper.
>
> We have responded to another reviewer concerning a similar question they had. Here is the response we provided.
>
> One of the main arguments for combining the two components (adaptive loss and multi-scale prior) under one unified framework is that they are synergetic.
> How do we explain this synergy? Compared to other transformer architectures (notably the baselines used), Scaleformer has more iterative steps: the sequential multi-scale operations. As is made clear from the example of e.g. RNNs, iterative computation tends to case more accumulation of errors, which will behave like outliers for the purpose of this loss. As a result, the process that leads to the need for the two components can be expressed as:
>
> (1) The multi-scale architecture is beneficial for performance due to being a useful structural prior for time series data.
>
> (2) The multi-scale architecture however relies on sequential computation that increases the likelihood of explosive error accumulation.
>
> (3) The adaptive loss serves to mitigate this issue, leading to more stable learning and better performance.
>
> As we recognize this is an important point that we should have made clearer in the submission, we added a section (Section E) in the updated appendix and references in the main paper highlighting the points above.
>
> 2. Similarity with the U-Net structure.
>
> We thank the reviewer for pointing out this interesting work. We have cited it as related work in the updated submission.
> This is an interesting observation and we agree with the reviewer that one can consider the downsampling part of our model as a fixed encoder followed by the core transformer and also replace the upsampling with a learnable decoder. However, directly comparing, despite the simplicity of our model, there are other core differences, such as having loss in every scale in our case which is not happening in the U-Net structure, or the limitation causes by using convolutional layers and residual connections in U-Net comparing with the Transformer architecture, etc.
>
> In a sense, the U-Net inspired model suggested by the authors is interesting, but also different from our submission conceptually for the reasons outlined above. We are happy to engage in more discussion on this point.
>
> 3. Normalizing the mean versus mean and standard deviation.
>
> We thank the reviewer for asking this question. We have chosen to answer this in our global response to reviewers (coming soon).
>
> 4. Limited gains for Autoformer from the normalization alone.
>
> This is an interesting question. The reason is that Autoformer benefits from an inner ìseries decompositioníí module which acts as the normalization by nature. Indeed one benefit of our proposed framework is a simple solution to bring the benefits of these specified designs to other baselines which significantly reduces the gap between for example Informer and Autoformer.
> We thank the reviewer for pointing out this interesting aspect. Our hypothesis for this is that AutoFormer already benefits from an internal component that reduces internal distribution shift: the series decomposition module. We will add a paragraph on this in the appendix to address this.
>
> We highly appreciate your time for reading our response.

---

> > ### Author Response · Authors · 2022-11-19
> > **More discussion on the mean versus mean and standard deviation**
> >
> > We added more discussion of normalization on both **section L in appendix** of updated submission and the general response.

---

### Official Review · Reviewer_VvHJ · 2022-10-25

**Confidence:** 4
**Correctness:** 3
**Technical Novelty And Significance:** 3
**Empirical Novelty And Significance:** 3
**Recommendation:** 6

**Clarity, Quality, Novelty And Reproducibility:**

* The paper is well written and is very thorough with the problem formulations.
* Most of the experiments are performed on standard time-series datasets. The framework needs to be validated on datasets like samples with chaotic time series, seasonality, and the time-series changes abruptly based on external factors.
* Since the authors released the Github code; hence all the experiments are reproducible.

**Strength And Weaknesses:**

Strengths:
* The novelty of the paper is interesting,
* The idea of time-series forecasting for multiple scales, by minimizing the distribution shifts between scales and windows using cross-scale normalization, is an interesting contribution to the different domains: Neuroscience, Stock Market, Weather Forecasting, etc.
* The paper is well written and is very thorough with the problem formulations (Both Architecture and its components are well explained with mathematical proofs).

Weaknesses:
* The Informer-MSA results are much closer to the actual time series compared to other methods in Figure 5; however, Fedformer and Autoformer display lower MSE and MAE in Tables 2 and 3. It is not clear to me why these models are performing poorly in the qualitative analysis compared to Informer-MSA.
* Figure 4 requires standard error bars for better comparison across the baseline models. Also, the authors could perform a statistical significance test to compare the results.
* While the authors focused more on comparing the model performances, whether authors have tried if we have frozen the layer of the Transformer and only train at the final layer? How do the frozen model effects approximate dynamic time-series forecasting?
* It will be interesting if authors can present the layer-wise representations across the models and how these models learn the temporal aspect while forecasting the time series.


**Summary Of The Paper:**

The paper proposes a transformer-based framework for time-series forecasting for multi-scale frequency components. The solution is inspired by two recent state-of-the-art transformer architectures used as backbones by iteratively refining a forecasted time series at increasingly fine-grained scales and introducing a normalization scheme that minimizes distribution shifts between scales.

**Summary Of The Review:**

Overall, the authors propose a novel framework Transformer-based model for time series forecasting for multiple scales. However, the paper requires much discussion on why one model is superior to the other and interprets the layer-wise representations.

---

> ### Author Response · Authors · 2022-11-16
> **Response to VvHJ**
>
> We thank the reviewer for your positive, insightful and valuable comments and suggestions which are very crucial for improving the quality of our manuscript.
>
> 1. Qualitative analysis.
>
> We thank the reviewer for their comment on the qualitative samples. We agree with your assessment that the qualitative samples in Figure 5 show FedFormer underperforming. This is due to two main issues with qualitative samples in general:
> - While FedFormer and newer models perform well on when averaged over the test dataset, this hides a large inter sample variation. Every model has a significant number of failure modes. As the samples we showcase are picked at random, they might show one model performing better than its average. This example was meant to be illustrative of the type of benefits captured by the multi-scale prior, whereas the empirical demonstration of improved performance relies on the results table themselves. To mitigate this issue, we have added additional qualitative results in Figure 9 of appendix .
> - It is often difficult to directly link qualitative, human-perceived performance with the resulting metrics. As an example to highlight this, average value baselines perform reasonably well on many datasets, but (being constant functions) are immediately identifiable by humans. Similarly, a sample that correctly identifies the frequency, but not the phase of a signal might result in very poor MSE. We wish to highlight that Figure 5 aims mostly to show the scaleformer-based models capture trends (and other human-relevant) information better than their baselines. We have added details in the main paper (section 4.3) to emphasize this.
>
> 2. Error bars and statistical test.
>
> We thank the reviewer for suggesting this: it helps us strengthen the submission. We have added error bars to the Figure 5 in the updated submission. We will also add the results of the statistical tests (coming soon).
>
> 3. Other points.
>
> We are still working on the best way to address the reviewers other comments. We will provide an update on this soon.
>
> We highly appreciate your time to read our response.

---

> > ### Author Response · Authors · 2022-11-19
> > **More discussion on synthetic dataset (e.g. chaotic dataset)**
> >
> > We added the **section M of appendix**  in the updated paper to further discuss our mutli-scale framework on synthetic dataset.
> >
> > We generate a chaotic dataset based on the Mackey–Glass equations (Mackey & Glass, 1977). Concretely, we use the following equation:
> > \begin{equation}
> > \frac{dx}{dt} = \frac{0.2\times x(t - \tau)}{1 + x(t - \tau)^{10}} - (0.1\times x(t)),
> > \end{equation}
> > where we follow  L  ́opez-Caraballo et al. (2016); Farmer et al. (1983) and we also consider $x(0)=1.2$. We create three series of length 10k. First we create a series by only using the above equation and considering $\tau=18$ where the series has a chaotic behaviour when $\tau\geq17$ . We also provide two additional series with $\tau=12$ and $\tau=9$, and also with added seasonal and trend components. Our synthetic dataset is a combination of these 3 series (See Figure 11 for an example). We compare our method using the new dataset on both Informer and Autoformer in Table 12, showing our proposed framework significantly improves the performance over the baselines.

---

> > ### Author Response · Authors · 2022-11-19
> > **Adding the results of statistical tests.**
> >
> > We have added the section N of the appendix containing the results of two sets of statistical tests. In these, we perform Student's t-tests to determine, for the FedFormer architecture the following:
> > - Whether the adaptive loss contributes to improving results when added.
> > - Whether the multi-scale architecture contributes to improving results when added.
> >
> > The results, detailed in the corresponding section, are in line with Figure 4 of the main paper. In almost all cases, the multi-scale architecture brings benefits that are statisticall significant (p < 0.05). The adaptive loss brings statistically significant benefits in most cases, but as Figure 4 indicates, has fewer measurable benefits for the exchange rate dataset.
> >
> > Note that the two settings evaluated are:
> > - FedFormer versus FedFormer-A
> > - FedFormer versus FedFormer-MS

---

### Official Review · Reviewer_N6mC · 2022-10-25

**Confidence:** 4
**Correctness:** 3
**Technical Novelty And Significance:** 2
**Empirical Novelty And Significance:** 3
**Recommendation:** 6

**Clarity, Quality, Novelty And Reproducibility:**

The technical contribution of this paper is incremental. Detailed implementation details are shared. The overall presentation could be improved with more clarification on the model description and training process.

**Strength And Weaknesses:**

Strengths
1. SOTA methods are compared in the paper.
2. Comprehensive experiments are conducted.

Weaknesses:
1. Technical contribution is incremental.

**Summary Of The Paper:**

In this paper, the authors propose a general multi-scale framework for time series forecasting.

**Summary Of The Review:**

In this paper, the authors proposed a  multi-scale framework that can apply directly to the existing time series forecasting framework. Detailed comments are listed below:
1. The technical contribution of this paper is a little bit incremental. The idea of leveraging multiple time resolutions of time series is not something new. The major contribution comes from cross-scale normalization.
2. How is the model trained with learnable parameters in the loss function? Also, how are alpha and c learned during the training?
3. Based on Figure 4, it seems that the adaptive loss doesn't have significant benefits until the multi-scale framework is used. Can the authors provide a move analysis and discussion?
4. It's observed that, in many cases (Tables 2 and 3), a single-scale model with mean normalization performs significantly better than the multi-scale version without normalization. Does that mean a simple mean normalization is more important than the multi-scale information in the time series? Can the authors provide more insight analysis?

---

> ### Author Response · Authors · 2022-11-16
> **Response to reviewer N6mC**
>
> We thank the reviewer for your positive, insightful and valuable comments and suggestions which are very crucial for improving the quality of our manuscript.
>
> 1. Question about the adaptive loss.
>
> All the parameters ($\alpha$, $c$) of the loss function are end-to-end trainable and learned by back-propagation during training.
>
> 2. Interplay between adaptive loss and multi-scale architecture.
>
> We agree with the reviewer concerning this finding. This is one of the main arguments for combining the two components under one unified framework: they are synergetic.
> How do we explain this synergy? Compared to other transformer architectures (notably the baselines used), Scaleformer has more iterative steps: the sequential multi-scale operations. As is made clear from the example of e.g. RNNs, iterative computation tends to case more accumulation of errors, which will behave like outliers for the purpose of this loss. As a result, the process that leads to the need for the two components can be expressed as:
>
> (1) The multi-scale architecture is beneficial for performance due to being a useful structural prior for time series data.
>
> (2) The multi-scale architecture however relies on sequential computation that increases the likelihood of explosive error accumulation.
>
> (3) The adaptive loss serves to mitigate this issue, leading to more stable learning and better performance.
>
> As we recognize this is an important point that we should have made clearer in the submission, we added section E in the updated appendix and reference in the main paper highlighting the points above.
>
> 3. Single-scale model with mean-normalization.
>
> We thank the reviewer for pointing this out: we realize that we should highlight and discuss it more in the submission.  We agree with the assessment: indeed often a single-scale model with mean normalization performs significantly better than the multi-scale version without normalization. The reason for this is that normalization as per our proposed scheme is targeted at all forms of internal distribution shift, not only those induced by the multi-scale architecture. In our submission, we make the case for the multi-scale prior as a natural prior to add to transformers. From empirical observations, we found that such a prior requires adapting normalization. When investigating means of normalizing, we observed additional benefits to non-multiscale architectures as well. This means that they also suffer from other forms of distribution shift: we attribute it in the paper to e.g. shifts between lookback and forecast distributions.
>
> We also highlighted this important point in a dedicated section (section F) in the appendix of the updated submission.
>
> Thanks a lot reviewer for your valuable time to read this response.

---

### Official Review · Reviewer_UakB · 2022-10-25

**Confidence:** 4
**Correctness:** 3
**Technical Novelty And Significance:** 2
**Empirical Novelty And Significance:** 2
**Recommendation:** 5

**Clarity, Quality, Novelty And Reproducibility:**

The methodology is presented clearly and the empirical studies are thorough, but the motivation is not quite clear and the novelty is marginal. The work is expected to be easy to be reproduce as source code is provided.

**Strength And Weaknesses:**

Strengths:
+ The paper is well-organized and the essential ideas are easy to follow.
+ Extensive experiments are conducted with impressive improvement in main results.

Weakness
- The introduction of multi-scale structure is not well motivated. It would be nicer if a good motivating example of using multi-scale forecasting is provided other than the ablation study.
- The reason of using average only in cross-scale normalization is not well justified. Why would it be able to address distribution shift without standardization as well?
- The adaptive loss is introduced for outliers but there is no such demonstration. And the improvement of adaptive loss seems marginal in the ablation study (figure 4)

**Summary Of The Paper:**

This paper presents a Transformer-based framework that iteratively forecast time series at different scales with shared weights. In particular, it proposes normalizing the downsampled time series to avoid distribution shift and using adaptive loss to deal with outliers. Experiments show the multi-scale framework can significantly improve the performance of various transformer-based forecasting models.

**Summary Of The Review:**

Overall I believe this paper make incremental contribution to forecasting community with a multi-scale framework. The empirical results are inspiring, however, for future research.

---

> ### Author Response · Authors · 2022-11-16
> **Response to reviewer UakB**
>
> We thank the reviewer for the insightful and valuable comments and suggestions. They are very valuable for improving the quality of our manuscript.
>
> 1. Motivating the multi-scale approach
>
> After discussing in detail, we agree with the reviewer concerning the motivation for the multi-scale approach. As this structural prior is crucial to understanding our submission, we have made the following modifications to the manuscript to clarify this point:
> - We have added a section in the appendix motivating the use of a multi-scale architecture (Section B of appendix ).
> - We have updated the manuscript to add a few citations mentioned above in the introduction, to create a stronger link with this.
>
> For your convenience, here is the motivating paragraph we have added, with citations attached:
>
> This section aims to provide more motivations for the use of a multi-scale architecture. Let us first consider the following classical example, highlighted in section 2 of Ferreira et al. (2006), corresponding to the monthly flows of the Fraser River from January of 1913 to December of 1990. As shown in the their corresponding plot, the annual averages are strongly inter-related, pointing to the fact that seasonality alone will not suffice to model the variations. In the context of the paper, this showcases a failure mode of an ARMA model, but this failing is more general: models that do not explicitly account for inter-scale dependencies will perform poorly on similar datasets.
> Different approaches have attempted to introduce multi-scale processing (Ferreira et al., 2006; Mozer, 1991) in ways that differ from our own approach. The multi-scale temporal structure for music composition is introduced in (Mozer, 1991). Ferreira et al. (2006) proposed a time series model with rich autocorrelation structures by coupling processes evolving at different levels of resolution through time.
> However, their base models are constrained to simple statistical models, e.g. Autoregressive models.
> To conclude, we note the following: (1) the approaches mentioned above have applied multi-scale modeling with success, and (2) we are the first work to explicitly consider a multi-scale prior by construction for transformers.
>
> Ferreira, Marco AR, David M. Higdon, Herbert KH Lee, and Mike West. "Multi-scale and hidden resolution time series models." Bayesian Analysis 1, no. 4 (2006): 947-967.
>
> Mozer, Michael C. "Induction of multiscale temporal structure." Advances in neural information processing systems 4 (1991).
>
> 2. Justifying limiting ourselves to mean-normalization (rather than mean- and standard deviation-normalization)
>
> We have chosen to answer this point in our global response to all reviewers. (Coming soon with more experimental results)
>
> 3.  Justification for the adaptive loss.
>
> Mathematically, the justification for the adaptive loss is as follows. Considering the $\xi$ term in equation 10 in our submission, the function is asymptotically close (but not equivalent due to the denominator) to $\xi^\alpha$. As a result, for outliers (for which $\xi$ will be large), the loss term will function as a $L^\alpha$ penalty on $\xi$, which will penalize outliers more for large $\alpha$. The converse of this is that we would expect a model trained with such a loss to learn lower values of $\alpha$ for settings with fewer outliers. We have added this justification to the section D of appendix.
>
> Currently we are also running experiments to validate this, and we will provide experimental results soon.
>
> We highly appreciate the reviewer's valuable time for reading this.

---

> > ### Author Response · Authors · 2022-11-19
> > **More discussion on adaptive loss justification**
> >
> > We added more discussion on adaptive loss justification on both **section M in appendix** of updated submission and the general response.

---

### Author Response · Authors · 2022-11-10
**Initial reply to all reviewers**

We thank all the reviewers for your valuable and insightful comments, which are very helpful to improve our work.

Broadly speaking, the comments fall under the following categories:
- Comments pertaining to the normalization method.
- Comments related to the use of the adaptive loss.
- Further analysis of results and learned representations.
- Other important points.

To provide the best answer to these comments will require additional experiments. We are currently running these, and will reply to each of you individually in the following days.

---

> ### Author Response · Authors · 2022-11-16
> **Follow-up to the initial response to all reviewers**
>
> We thank all the reviewers for their valuable and insightful comments. We are also grateful to the reviewers for their positive comments on our work.
>
> As mentioned in our initial response to all reviewers, broadly speaking, the reviewers’ questions fall under the following categories:
>
> - Relating to the normalization method.
> - Relating to the adaptive loss.
> - Analysis of the results or learned representations.
> - Other important points not mentioned above.
>
> We have chosen to answer the first point here, as it occurs frequently in the different reviews. The other points are addressed in our individual responses to each reviewer.
>
> Explaining the motivation for our choice of normalization method
> -----------------
> Two main issues are faced during training. On one hand, (internal) distribution shifts hinder performance, as demonstrated in the experiments in the main paper (Table 2). This can be mitigated by normalizing the inputs. On the other hand, normalizing the inputs means less information is available to the model when making forecasts. Indeed, information about mean and standard deviation of the input data is not available when processing this input.
>
> When running initial experiments for our model, we considered both mean-only, and mean-and-standard-deviation normalization. The first option performed better on average. We have added Table 11 to the appendix of the updated paper. It showcases the results of these experiments.
>
> Returning to the previous point about information loss, we believe that the results in Table 11 can be explained by the fact that normalizing by the mean only means more information about the distribution of inputs/representations is available to the model. As normalization is still performed, and empirically is found to be sufficient to prevent excessive distribution shifts, this setting seems more adequate.
>
> We are happy to discuss this, and other points, with the reviewers.

---

> > ### Author Response · Authors · 2022-11-18
> > **More discussion on cross-scale normalization**
> >
> > To further discuss the cross-scale normalization. we added **section L in appendix** of updated submission. Here are the main paragraphs:
> >
> > The choice of normalization approach is essential to our method, due to the aforementioned distribution shifts. In this section, we showcase multiple experiments that were done to study the impact of different ways of normalizing inputs. In Table 11, we compare using two forms of normalization with Informer-MSA as the baseline: mean-only versus normalization using both mean and standard deviation. We consider that adding normalization to our method is a trade-off between two issues faced during training. One one hand, (internal) distribution shifts hinder performance, as demonstrated in the experiments (See Table 2). On the other hand, normalizing the internal representations results in a form of information loss (during the processing of the input): we lose information about mean
> > and standard deviation of the input data. We believe that the reason mean-normalization outperforms mean and standard deviation normalization is because it results in a better trade-off, losing less information while still managing to address the internal distribution shift.
> >
> > Fig. 10 shows the different dataset distributions. As we can see, mean normalization works better for datasets such as electricity and traffic with almost unimodal distributions. For datasets such as exchange-rate and to some extent weather, which have multi-modal distributions, we find that mean and standandard deviation works better. Our hypothesis is that mean-normalization is too
> > simple an approach for more complex input distributions. We note that, in all cases, **either form of normalization results in strong improvements compared to the absence of normalization**.

---

> > ### Author Response · Authors · 2022-11-19
> > **More discussions on adaptive loss**
> >
> > To further discuss the effect of adaptive loss. we added **section M in appendix** of updated submission. Here are the main paragraphs:
> >
> > To further evaluate our proposed framework and adaptive loss function, we generate a new dataset based on the Mackey–Glass equations (Mackey & Glass, 1977) Concretely, we use the following equation:
> > \begin{equation}
> > \frac{dx}{dt} = \frac{0.2\times x(t - \tau)}{1 + x(t - \tau)^{10}} - (0.1\times x(t)),
> > \end{equation}
> > where we follow  L  ́opez-Caraballo et al. (2016); Farmer et al. (1983) and we also consider $x(0)=1.2$. We create three series of length 10k. First we create a series by only using the above equation and considering $\tau=18$ where the series has a chaotic behaviour when $\tau\geq17$ . We also provide two additional series with $\tau=12$ and $\tau=9$, and also with added seasonal and trend components. Our synthetic dataset is a combination of these 3 series (See Figure 11 for an example). We compare our method using the new dataset on both Informer and Autoformer in Table 12, showing our proposed framework significantly improves the performance over the baselines.
> >
> > In addition, to further analyze the effect of the adaptive loss on noisy datasets with outliers, we randomly replace a percentage of the training data with outliers by defining an outlier as a sample with a distance of larger than 50 times of standard deviation of the series from the median. Figure 11 shows the improvement of using adaptive loss on both Informer baseline and our multi-scale version of Informer. Adaptive loss increases the performance up to roughly 70% when there is a noisy input with extreme outliers ( 5% of data) while it gets comparable results for a clean dataset, demonstrating it as a strong candidate to replace MSE loss in the time series forecasting tasks.

---

### Decision · Program_Chairs · 2023-01-20

**Decision:**

Accept: poster

**Justification For Why Not Higher Score:**

Unless there is a mechanism in the review process that allows us to review the revised paper before publication, I don’t feel comfortable of recommending for spotlight acceptance.


**Justification For Why Not Lower Score:**

There is one vote for rejection (5), but in fact the reviewer also praises this work as indicated in the strengths. The weaknesses are more related to presentation and further elaboration which can be fixed in the revision. In other words, they are not fatal.


**Metareview: Summary, Strengths And Weaknesses:**

Deep learning models such as recurrent neural networks have been applied to time series forecasting. With the surge of research interests in transformer-based methods in recent years, it is a natural next step to apply them to time series forecasting as well. Nevertheless, a number of subtle issues have to be addressed. Instead of proposing yet another model, this paper proposes a multi-scale framework that can be applied to existing transformer-based time series forecasting models. Good results are obtained in the experiments. In addition, since it does not introduce much computational demand, we are convinced that the proposed framework will see adoption by practitioners to further enhance the performance of their preferred choice of SOTA transformer-based models. Nevertheless, the paper in its original form does have much room for improvement. We thank the authors for responding to our comments during the discussion period. We believe many of the concerns, such as those related to the motivation and presentation of the paper, can be addressed before publication. We highly recommend the authors to significantly revise their paper by incorporating the clarifications and changes made during the discussion period and also making further changes as deemed necessary.


**Note From Pc:**

if the above contains the word "oral" or "spotlight" please see: "oral" presentation means -> notable-top-5% and "spotlight" means -> notable-top-25%. As stated in our emails, we are disassociating presentation type from AC recommendations